# Maximizing carbon sequestration potential in Chinese forests through optimal management

Zhen Yu [1,2] ✉, Shirong Liu [2] ✉, Haikui Li[3], Jingjing Liang[4], Weiguo Liu [5], Shilong Piao [6], Hanqin Tian [7], Guoyi Zhou [1], Chaoqun Lu [8], Weibin You[9], Pengsen Sun[2], Yanli Dong[1], Stephen Sitch [10] & Evgenios Agathokleous [1]

Forest carbon sequestration capacity in China remains uncertain due to underrepresented tree demographic dynamics and overlooked of harvest impacts. In this study, we employ a process-based biogeochemical model to make projections by using national forest inventories, covering approximately 415,000 permanent plots, revealing an expansion in biomass carbon stock by 13.6 ± 1.5 Pg C from 2020 to 2100, with additional sink through augmentation of wood product pool (0.6-2.0 Pg C) and spatiotemporal optimization of forest management (2.3 ± 0.03 Pg C). We find that statistical model might cause large bias in long-term projection due to underrepresentation or neglect of wood harvest and forest demographic changes. Remarkably, disregarding the repercussions of harvesting on forest age can result in a premature shift in the timing of the carbon sink peak by 1−3 decades. Our findings emphasize the pressing necessity for the swift implementation of optimal forest management strategies for carbon sequestration enhancement.

Forests play a central role in achieving the UN Sustainable Development Goals (SDGs), including sustainable use of terrestrial ecosystems, combating desertification, reversing land degradation, and halting biodiversity loss[1,2]. With mounting interests in forest restoration to mitigate global climate change[3], maintaining and enhancing forest carbon sequestration have been increasingly prioritized[4]. China, the top-ranked country worldwide in planted forest area[5], is expected to be a substantial and persistent carbon sink in the coming decades[6]. However, reported estimates of forest carbon potential in China remain inconsistent due to large uncertainties caused by limited long-term reliable observational data, different accounting methods, and varied forest management plans and strategies[6–14] (Fig. 1). For example, China has pledged to expand its planted forest area by 49.5 million hectares (Mha) from 2021 for land carbon sink enhancement[15], yet the appropriate forestation species adapting to climate change are not identified. The timeframe for achieving the anticipated carbon benefits will predominantly hinge on several key factors: the judicious selection of tree species, the evolving patterns of wood harvesting practices, the

[1]Key Laboratory of Ecosystem Carbon Source and Sink, China Meteorological Administration (ECSS-CMA), School of Ecology and Applied Meteorology, Nanjing University of Information Science and Technology, Nanjing 210044, China. [2]Key Laboratory of Forest Ecology and Environment, China's National Forestry and Grassland Administration, Ecology and Nature Conservation Institute, Chinese Academy of Forestry, 100091 Beijing, China. [3]Key Laboratory of Forest Management and Growth Modelling, China's National Forestry and Grassland Administration, Research Institute of Forest Resource Information Techniques, Chinese Academy of Forestry, 100091 Beijing, China. [4]Forest Advanced Computing and Artificial Intelligence Laboratory (FACAI), Department of Forestry and Natural Resources, Purdue University, West Lafayette, IN 47907, USA. [5]College of Forestry, Northwest agriculture and Forestry University, Yangling 712100, China. [6]Sino-French Institute for Earth System Science, College of Urban and Environmental Sciences, Peking University, 100871 Beijing, China. [7]Schiller Institute for Integrated Science and Society, Department of Earth and Environmental Sciences, Boston College, Chestnut Hill, Massachusetts, MA 02467, USA. [8]Department of Ecology, Evolution, and Organismal Biology, Iowa State University, Ames, IA 50011, USA. [9]College of Forestry, Fujian Agriculture and Forestry University, Fuzhou 350002, China. [10]College of Life and Environmental Sciences, University of Exeter, Exeter, UK. ✉e-mail: zyu@nuist.edu.cn; liusr@caf.ac.cn

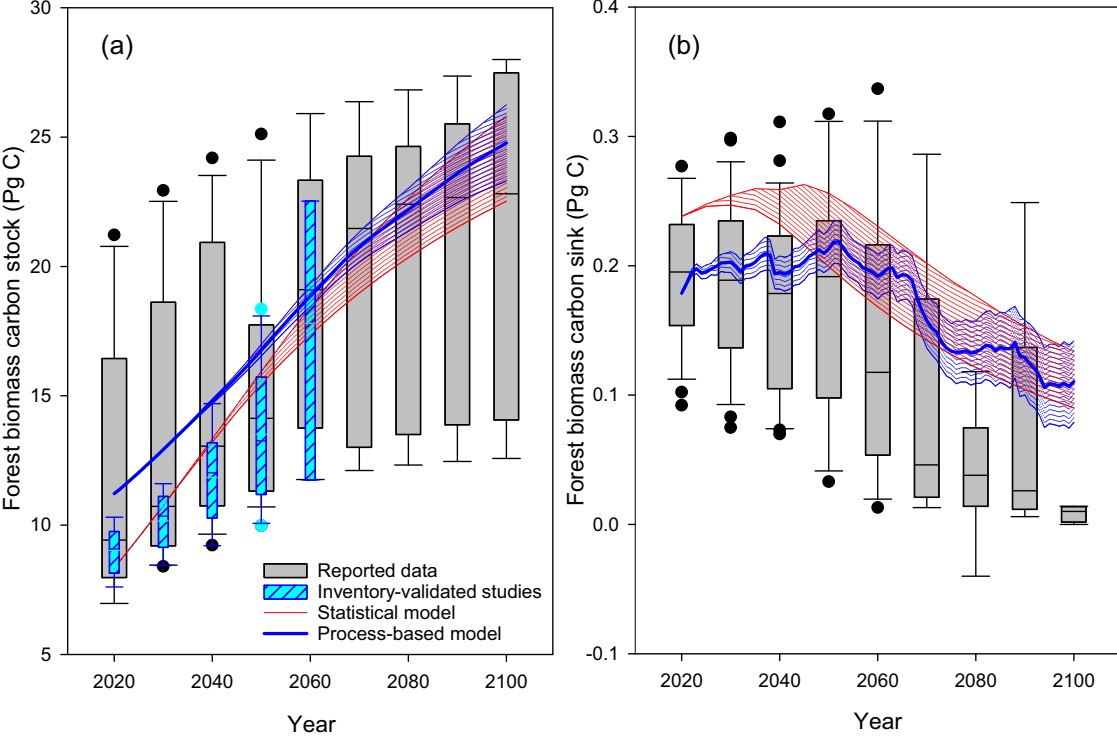

**Fig. 1 | Forest biomass carbon stock and sink in China.** Boxplot shows the median (center line), and the lower (10th) and upper (90th) quartiles; outliers shown as individual data points. Gray box: the pooled estimations from publications; cyan box in panel a: the reported carbon stock using models validated by historical forest inventory data; red shaded area in panels **a** and **b**: the range of carbon stocks and sinks derived from statistical model under the baseline (47% tree survival rate) and the improved management scenarios (85% tree survival rate) in this study; blue shaded area in panels **a** and **b**: the stock and sink derived from process-based simulations using DLEM under different tree survival rates (i.e. 47% and 85%) and Shared Socioeconomic Pathways scenarios in this study. The improved management practices (i.e. tree replacement and harvest rotation length extension) and wood product pool were not considered or included.

structural composition of forests, and the dynamics of tree demographics (i.e. forest age structure).

However, the current research gap mainly lies in the lack of collective considerations of the above key factors, which further constrains previous assessments of China's forests' carbon-holding capacity. Therefore, this study aims to: 1) address these challenges in an integrated manner for more accurate projection of forest biomass carbon potential; and 2) assess the improved management practices to enhance carbon potential. Here, we quantified forest biomass carbon stock (excluding dead wood, litter, and soil carbon) dynamics across China from 2020 to 2100, leveraging detailed tree-level information derived from the National Forest Inventory (NFI) survey data encompassing 18 million+ tree-records collected in more than 415,000 permanent sample plots from 1999 to 2018 (see *National forest inventory (NFI) data* in Methods, Supplementary Fig. S1). These forest inventory data, which provided validated measures of the initial carbon stock at the individual tree level (i.e. in 2018), enabled us to estimate forest biomass carbon stock and sink with a greatly improved accuracy. We employed a combination of both statistical and process-based biogeochemical models for carbon stock and sink projection. Specifically, in the simulations using process-based model (i.e. DLEM), forest management (i.e. wood harvest, forest expansion) was represented for accurate projection of carbon dynamic trajectory. Besides, the improved management practices were further considered, including the replacement of inappropriate tree species by indigenous species and reduction of the harvesting intensity by postponing the harvesting time (see Methods). Additionally, optimal managements include the improved management practices producing the largest carbon sink. Moreover, for DLEM simulations, we designed four groups of experiments, in which the first group is to quantify the carbon dynamics

without improved management practices, and the second group is to quantify the additional carbon stock and sink from improved forest managements, i.e., wood harvest extension and tree replacement. While in comparison, the third group experiments were used to quantify the bias introduced by neglecting wood harvest, and the fourth group was used to quantify the sink from existing and new forests. Note that the results reported are all derived from group 1 experiments, unless otherwise indicated.

## Results and discussion

We optimized wood harvest based on forest age and the dynamic patterns of forest demography. More specifically, instead of demand-driven carbon removal such as Land-Use Harmonization (LUH2), harvesting was determined internally based on tree maturity status specified by tree species and age in our simulations. Moreover, forest age and demography were altered by harvesting practice which, in turn, further affected the forest growth. These mechanisms improved the carbon removal and accumulation processes in the projection of future biomass carbon. Former studies[16,17] revealed that forest demography played a vital role in terrestrial carbon sink, and the carbon accumulation rates were extremely divergent during forest regrowth across the globe. Consequently, we enhance the accuracy of estimates by improving forest growth and regrowth processes in model simulations.

Aside from harvesting and the subsequent impacts on forest demography and growth, future forestation species will also greatly determine the carbon potential. In this study, the tree species selected for future forestation were appropriately determined using observation-based, improved habitat suitability maps (see Methods). We found that, previous studies, with initial carbon stock validated by

inventory data[7,12,18,19], are less diverged in reported carbon stocks than the results lacking validations (see cyan box and gray box in 2020 in Fig. 1a). Studies with abnormally high initial forest carbon stock are prone to report lower carbon accumulation potential (e.g. Ju et al.[14] in Supplementary Table S1) as the carbon saturation is ecologically and biologically predefined in model simulations (Supplementary Table S1, Supplementary Figs. S2–S5). Thus, we first calculated the biomass for each tree species by the continuous biomass expansion factor method (Supplementary Table S2) and converted it to biomass carbon based on species-specific parameters (Supplementary Table S3) using the forest plots surveyed in the 9th NFI (2014–2018) (Supplementary Table S4). The biomass carbon stocks of each tree species were further used as the reference to validate the process-based model (i.e. DLEM) simulations in 2018 (Supplementary Fig. S6). We also modeled forest carbon stock changes in existing and new forests under different scenarios of tree survival rates for both statistical and process-based models from 2020 to 2100, while wood harvest, rising $CO_2$ concentrations, and climate change were further considered for process-based simulations (Method, Supplementary Table S5). For example, unlike the vast majority of the previous studies[7,9,18–20], which assumed that all new forestations were successful, we consider forestation failures by constraining tree survival rates between harsh conditions (e.g. arid and semi-arid regions) and the expected criteria designed by the State Forestry Administration. Specifically, for new planted forests, simulations were performed assuming a baseline tree survival rate (47%, derived from the 9th NFI in the arid and semi-arid region of Northwest China) and a scenario with elevated tree survival rate (85%, the threshold above which land qualifies for forestation as set by the State Forestry Administration of China) under improved forest management (see *Future forest expansion derived from official forestation plan* in Methods). The tree species adopted for forestation were based on habitat suitability maps with the dual considerations of future climate and the dominant species appropriate for each province (see Methods). Besides, the $CO_2$ concentrations and climate change forcing data were obtained from CMIP6 for each of the four Shared Socio-economic Pathways (i.e. SSP1-2.6, SSP2-4.5, SSP3-7.0, and SSP5-8.5).

In previous reports, projections of forest carbon stocks show greater uncertainties in later periods (Fig. 1a). Despite these studies calibrated and validated by NFI data help to converge the estimations (cyan box in Fig. 1a), the carbon stock uncertainties (indicated by the ratio of standard deviation and average) rise rapidly from 9% in 2020 to 25% in 2060 in previous studies, manifesting the importance of reducing such divergence. Our improved estimates, by exploring carbon dynamics under scenarios close to realistic conditions based on Chinese forestry plan with data-constrained by the NFI, agree well between the two methods used (i.e. statistical and process-based models), showing that forest biomass carbon stock will increase to 21.6–24.3 Pg C in 2100 (Fig. 1a). Specifically, the results of DLEM (i.e. process-based model) simulations were derived from group 1 experiments (see Supplementary Table S5), indicating an increase of 13.6 ± 1.5 Pg C from 2020 to 2100. This implies a sink potential of 11.2–14.8 Pg C to be sequestrated, and an additional of 1.0–1.1 Pg C that can be further fixed if tree survival rate increases from the current status (47%) to 85% for new planted forests. By pooling results from all scenarios with different tree survival rates and SSPs (i.e., Supplementary Experiments S1–S8 in Supplementary Table S5), the simulated forest biomass carbon sink will be 0.203 and 0.192 Pg C yr$^{-1}$ in 2030 and 2060 (Fig. 1b), which is 2.0–15.3% higher than a former estimate at 0.176–0.189 Pg C yr$^{-1}$[6]. This will offset 5.4–7.8% and 4.6–8.5% of the projected peak fossil fuel $CO_2$ emission in 2030 and 2060 in China, respectively[6,21].

In this study, the conspicuous difference was observed in biomass carbon sinks of the period 2020–2050, in which the magnitude derived from statistical model was 19% higher than process-based model (i.e. DLEM, see red and blue color in Fig. 1, which was derived

from experiments of simulation group 1). We hypothesized that this difference could be due to the impacts of wood harvest, rising $CO_2$, and climate change, which were not explicitly considered in the statistical model. To test the assumption, we further performed factorial simulations using DLEM with exclusion of the impacts from rising $CO_2$, climate change, or wood harvest (experiments in simulation group 3, see Supplementary Table S5 and Methods), respectively. The results revealed that the exclusion of wood harvest will help narrow the carbon sink difference between the two approaches down to 1.2% for the period 2020–2050 (0.001 Pg C yr$^{-1}$, Supplementary Fig. S7). Yet, the exclusion of rising $CO_2$ and climate change further enlarges the difference. Thus, the simulations using process-based model (experiments in simulation group 1, see Supplementary Table S5) are more realistic by taking into account of wood harvest impacts. The wood harvest was rarely considered in recent publications (since 2020), in which the sink covering the period 2020–2060 was reported at 0.21 ± 0.034 Pg C yr$^{-1}$[6,9,20,22,23] – i.e. about 4.5% higher than our estimations. In this study, we deliberately and proactively include the wood harvest as this allows to sustain forest regrowth associated with sink size capacity. Moreover, this follows the routine forest management for timber production.

Quantifying the wood product pool is challenging as the harvestable carbon is growth-dependent and further complicated by the greatly varied, species-specific rotation length ranging from 11 to 101 years[24]. What makes the estimates more complex is that the wood harvest will not only remove carbon from live biomass pools, but will also alter the age structure, biomass growth, and the forest responses to environmental changes (e.g. rising $CO_2$, climate change)[16]. Here, to improve estimates on carbon harvest impacts, we tracked the age dynamics of each tree species by considering natural growth, wood harvest, and tree mortality using over 50,000 forest plots surveyed in the 9th NFI (see Methods). By doing so, we could incorporate the age-related impacts of forest harvest on tree growth into biogeochemical cycle modeling. In this study, wood harvest was further constrained by official policies (i.e. harvest performed in timber forests only) and species-specific harvest age (see Methods, Supplementary Table S6). The harvested wood was diverted into wood products with varying decaying rates. The annual carbon removal by wood harvesting was at 55 Tg yr$^{-1}$ in our DLEM simulations from 2000–2020, which is in the range of 40–100 Tg yr$^{-1}$ that was reported in previous studies[5,25]. We estimated that wood product pool will reach 1.9 ± 0.1 Pg C in 2100 if the ratios of short-, mid-, and long-live wood products (i.e. paper/paperboard, wood-based panels, and sawn wood, respectively) remain the same as in 2020. In comparison, when no harvesting is implemented, the carbon stored in live biomass will be higher, at ~5.9 Pg C in 2100 (plus 0.2 Pg C remains in wood product pool due to a lack of new harvest wood input since 2020). This indicates a faster carbon turnover rate in current wood product pool than it in the live biomass in China. Although challenging, the long-live products (e.g. sawn wood) should be encouraged to increase carbon residence in the biosphere, which, however, are market-dependent, policy-oriented, and promoted by wood technological innovation as well for achieving carbon neutrality.

## Spatial distribution of projected carbon stock and sink
The process-based model is advantageous in describing the spatial pattern of carbon stock dynamic, which can hardly be captured in statistical modeling[26,27]. In this study, we forced the DLEM model using land use and cover change (LUCC) data intensively validated both spatially and temporally during the historical period[28,29]. We further extended the LUCC data from 2020 to 2050 assuming forestation activities deployed in each province and by year, strictly following official forestation plans (see Method). Moreover, we separated forests into timber and non-timber, because there are harvest-free forests (enforced by policy), such as forests in natural reserves and forest for

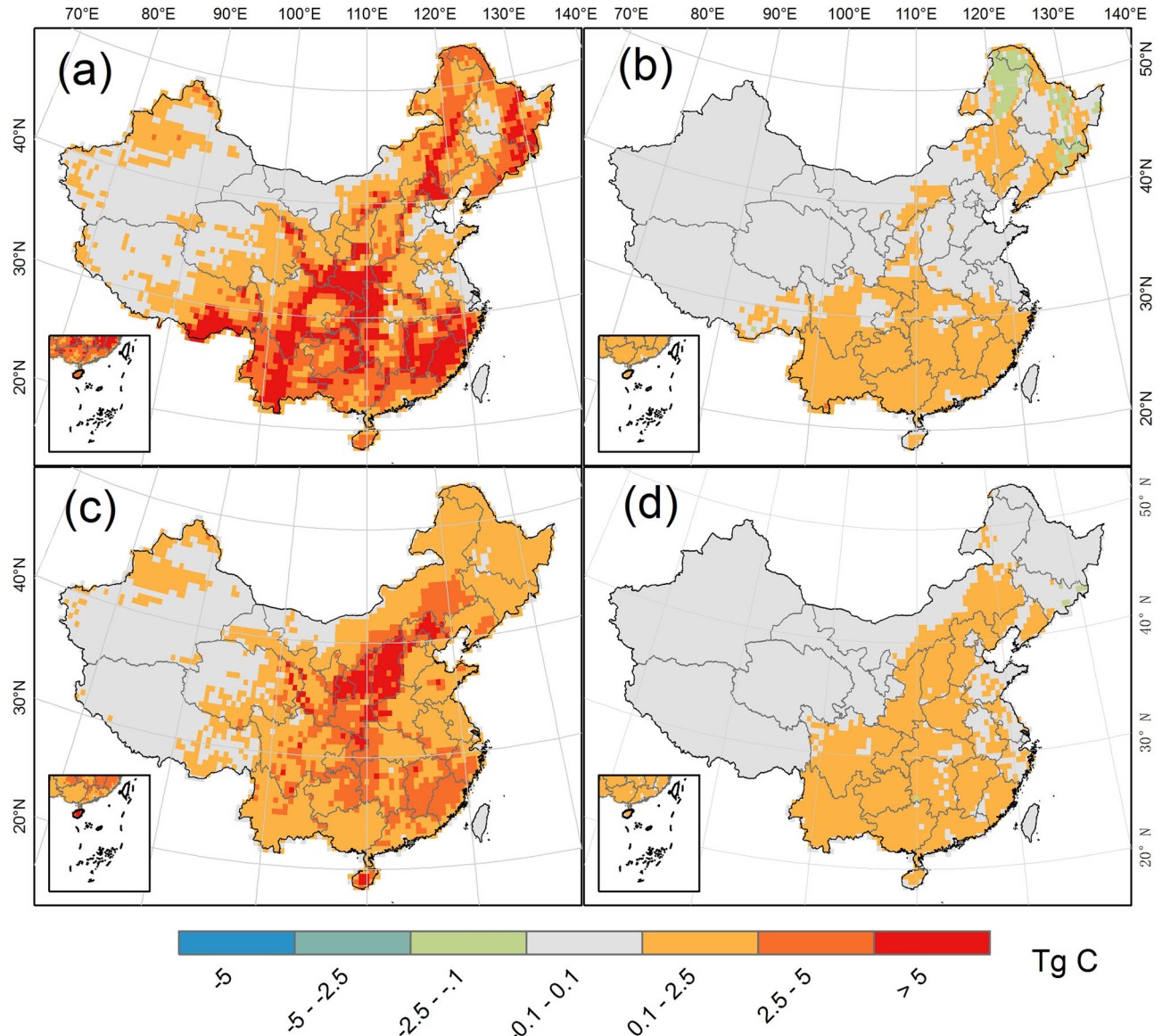

**Fig. 2 | The modeled biomass carbon stock change in China's forests from 2020 to 2100.** Panels **a**–**d** indicate the accumulated changes of biomass carbon stock in non-timber natural forest, timber natural forest, non-timber planted forest, and timber planted forest, respectively; unit: Tg C (1 Tg = 0.001 Pg). The species-specific areas of timber forest were obtained from the China Forest Resource Report 2014–2018. Results derived from process-based simulations under different tree survival rates (i.e. 47% and 85%) and Shared Socioeconomic Pathways scenarios. The management practices (i.e. tree replacement and harvest rotation length extension) and wood product pool were not considered or included.

sand fixation and soil conservation (i.e. non-timber forests). Thus, we only applied forest harvesting in natural and planted timber forests.

Process-based simulations reveal that the biomass carbon stock increment will be mainly derived from non-timber forests, including natural and planted forests. The natural forests comprise the majority of the carbon sink (57.7%) during the study period, followed by non-timber planted forests (34.2%) (Fig. 2). By comparing the simulations in group 1 and 4, we found that the existing and new forests contributed to 70.4 ± 3% and 29.6 ± 2% of the sink during the period of 2020–2100, respectively. It is noticeable that both natural and planted timber forests merely contributed to a lower proportion of the carbon stock increment (8.1%, Fig. 2). This is expected because carbon is routinely removed from timber forests, leading to a relatively constant carbon stock unless the traditional timber forest management are altered (Fig. 2).

Predicted by process-based modeling, the average biomass carbon sink would be 0.201 and 0.143 Pg C yr$^{-1}$ during the periods

2020–2060 and 2060–2100, respectively, resulting in an average sink of 0.172 ± 0.017 Tg C yr$^{-1}$ for the period 2020–2100 (Figs. 1 and 2). Although this represents approximately 7.2% of the global forest biomass sink, its size is 134.3% greater than that of the current U.S. forest carbon sink[30,31] (i.e. about 1.9 times higher by per unite of forest area), and 4–10 times the size of the terrestrial carbon sink of Europe during 1990–2100 (0.017–0.038 Pg C yr$^{-1}$)[32]. The simulated sink is 43.8% larger than the average sink during 2020–2100 that is derived from pooled studies regardless of the publication date (Fig. 1b), mainly due to a smaller magnitude of reported carbon sinks after 2060. There are two major reasons for this underestimate in carbon sink previously reported for the post-2060 period. First, forest areas in previous reports were either much smaller than they are in reality (e.g. 142.79–159 Mha in 2020 from Zhou and Liu[13,33] vs 220.45 Mha in 2018 from 9$^{th}$ NFI and this study) or did not account for future forest expansion[14]. In comparison, our historical forest maps were rigorously calibrated and validated, and the new forests were established

according to official plans with the consideration of land competition and climate changes in forestation projections (see Methods). Second, the non-validated and abnormally high initial carbon stock (20.36–21.22 Pg C in 2020 from Ju[14] vs 11.2 Pg C from this study) might limit the carbon accumulation potential due to carbon saturation (Supplementary Table S1, Supplementary Figs. S2–S5). Thus, we argue that former studies might underestimate the forest carbon potential for the study period.

Similar to previous studies[9,22,34], our results derived from statistical modeling also reveal that the forest carbon sink will peak during the 2020s–2040s (Fig. 1b). However, the process-based model reveals that the sink will peak in the 2050 s (Supplementary Fig. S8). We hypothesized that the early sink peak previously reported was due to 1) neglecting harvesting impacts on forest age, and 2) biases stemming from assuming forest age increment fixed at 1 per year. Specifically, timber forests routinely harvested will remain relatively young to uptake carbon, which was ignored if harvesting was not considered in carbon stock/sink projections. Moreover, former studies often assumed that forest annual age accrual is fixed at 1 per year due to lack of tree demographic data. In reality, except for timber forests that are routinely harvested with fixed rotation years, the age accrual in non-timber forests is complex and affected by mortality, regeneration, and natural growth (Supplementary Text; Supplementary Fig. S9). We derived the age accrual information from the 50,000 plots, which helps to capture the forest age changes under various disturbances. To test the assumptions, we designed an additional simulation with exclusion of harvesting and forced forest age increase by 1 year, which resulted into similar carbon sink peaks in the 2020s–2040s (Supplementary Fig. S10). Thus, we confirmed that the peak of forest biomass carbon sink will be 1–3 decades later than expected, and forest harvesting and age increment should be interactively considered in carbon sink projections. Despite the later peak, forest biomass carbon will level off with aging trees, causing an inevitable decline in carbon sink (Fig. 1). The plummet in sink may challenge the carbon neutral goal since 2060 in China (Fig. 1b). However, forest harvesting in some of the non-timber forests might benefit from diverted carbon to wood product pool. Nonetheless, it is crucial to rationally determined harvesting intensity in non-timber forests by trading the removed biomass stock, the accrued forest product pool, and the boosted sink. Furthermore, practices should be limited to selective cutting and should avoid reserved forests that serve important purposes such as biodiversity conservation, water conservation, soil erosion prevention, and sand control.

We further examined the spatial pattern of the year of carbon sink peak for the period 2020–2100 (Fig. 3a). Generally, the early carbon sink peaks were found in the east of Tibet Plateau and the southern regions, where the forests are either older or fast-growing and more likely to mature soon. In comparison, the later carbon sink peaks were distributed in the north of northeast and the southwest regions. Notably, the magnitudes of the peak sink are large and their sensitivities are high along the dividing line of humid and arid area (known as 'Hu Huanyong Line' or 'Heihe-Tengchong Line', Fig. 3c, d). This is consistent with a former study suggesting that future forestry practice will be emphasized in the area along the Hu Line[35].

## Improved management for carbon potential enhancement

Wood harvest might introduce a large carbon uptake signal in atmospheric inversion models as the harvested carbon is likely transported and decays far away from the producing area[36]. Therefore, the carbon sink signal derived from eddy-flux and atmospheric inversion approaches might be high in timber-producing regions[37,38], which is typical in the southern and southeast China where the Fast-Growing and High-Yielding Timber Base Construction Program has been implemented targeting at timber production[35]. We quantified the carbon stored in the wood product pool and the additional carbon that

can be sequestrated by implementing appropriate forest management practices. Despite wood product may prolong the carbon residence out of the atmosphere, the carbon-holding capacity of this pool is highly dependent on the residence/turnover time of the wood products[39]. Thus, wood products were aggregated into three categories in this study, namely sawn wood, wood panels, and paper/paperboard (see Methods). The decay of carbon in wood products was calculated by applying the Tier 2 method proposed by IPCC[40]. Our estimations revealed that the carbon removed from timber forest would be 7.8 ± 0.3 Pg C (96.6 Tg C yr$^{-1}$) during the period 2020–2100 (Supplementary Fig. S11), while the wood product pool would reach 1.9 ± 0.1 Pg C in 2100, about 5.9 Pg C lower than the biomass carbon pool if remains non-harvested. Therefore, current wood products decay much faster than the turnover of living biomass carbon, implying that slowing down the turnover of harvested wood products might help to prolong carbon residence out of atmosphere. We found that the rapid decay of wood products was primarily attributed to the high proportion of short-lived wood products such as paper/paperboard (61.5% based on FAO data). We estimated that, if the ratio of long-lived wood products (i.e. sawn wood: wood panel: paper) continues to increase from 2020 to 2060, the wood product pool would be 75.3% (1.4 ± 0.08 Pg C) higher compared to the scenario where it remains unchanged from 2020, emphasizing the significance of promoting the use of long-lived wood products.

Model simulations were designed to explore the additional carbon sink from forest management practices (Supplementary Table S5), including the extension of harvest length in timber forests and the replacement of tree species in non-timber forests (Methods). We found that timber forests with intensive carbon removals could lead to net carbon emission, while extending harvest cycle at the key year (i.e. the year in which the carbon sink levels off) for a certain species or forest type helps to reduce logging intensity and facilitates retaining carbon in forests (Methods). Results show the carbon sink can be enhanced by 2.5–3.3% (4.3–5.6 Tg C yr$^{-1}$) for timber forests if wood harvest age is postponed by 5 years after the key year (Table 1). Besides, if the inappropriate tree species were replaced by local indigenous species in 2025 (-4.13 Mha, see Methods), an additional 0.1–0.6% (0.17–1.1 Tg C yr$^{-1}$) carbon sink can be attained (in total of 43.9–166.8 Tg C accumulated from 2020–2100, Table 1). However, delayed action of both practices will reduce the accumulated carbon sink by 11–17 Tg C per year of delay under the four Shared Socioeconomic Pathways (SSPs, Table 1). The two management practices barely contributed to additional carbon accumulation for the period of 2020–2100 if implemented with 15-year delay, highlighting the importance of immediate implementation of forest management practices regardless of the future emission scenario.

In conclusion, China's forests are poised to function as a substantial net carbon sink, sequestering an estimated of 172.3 ± 16.9 Tg C yr$^{-1}$ over the course of 2020–2100. Furthermore, through the adoption of optimal management practices (i.e. implementation of both tree replacement and wood harvest extension by 5 years after key year), an additional sink of 28.1 ± 0.4 Tg C yr$^{-1}$ (totaling 2.3 ± 0.03 Pg C) can be achieved. Our findings underscore the distinct role played by China's forests in influencing carbon sink/source dynamics, which diverges from Canada's managed forests, predicted to act as a carbon source ranging from 0.007 to 0.024 Pg C yr$^{-1}$ between 2010 and 2100[41]. However, China's impact aligns more closely with the global forest carbon sink attributed to phosphorus (0.115 Pg C yr$^{-1}$) or nitrogen (0.233 Pg C yr$^{-1}$) deposition during the period 2030–2100[42]. Furthermore, promoting the extended use of long-live wood products emerges as a promising avenue to bolster the carbon sink and enhance carbon residence time in land. Consequently, it is imperative to expedite the implementation of carbon sink enhancement practices. To sum up, this study comprehensively projected carbon stock, sink, and potential in China's forests from 2020 to 2100. The assessment is

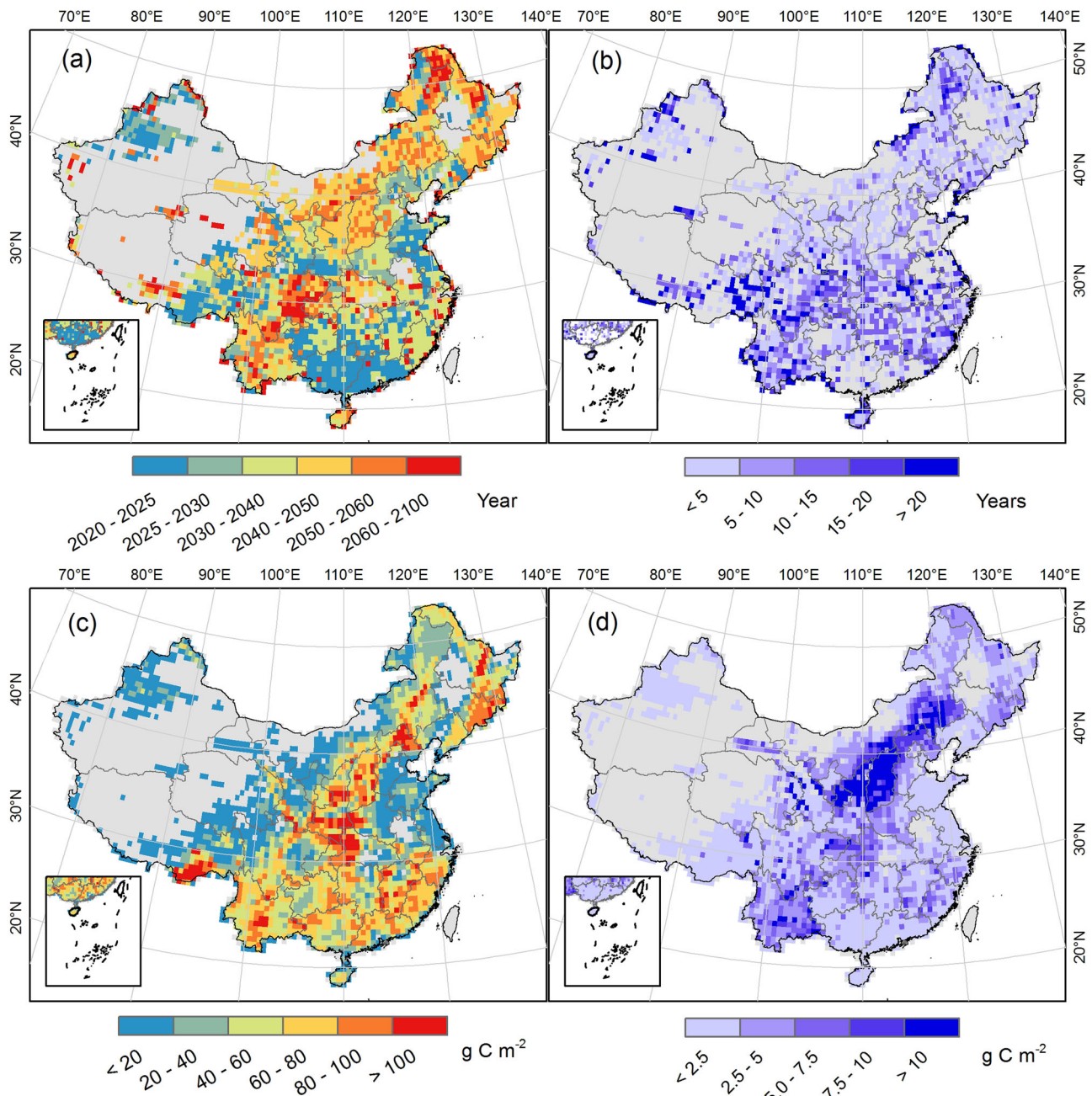

**Fig. 3 | The timing and magnitude of carbon sink peak in China.** Panel **a**: the year of carbon sink peak derived from the average of different Shared Socioeconomic Pathways (SSP) scenarios; Panel **b**: the standard deviation of the peak year; Panel **c**: the average magnitude of the carbon sink peak derived from different scenarios in g C m$^{-2}$ yr$^{-1}$; Panel **d**: the standard deviation of the carbon sink peak. Results derived from process-based simulations under different tree survival rates (i.e. 47% and 85%) and SSP scenarios. The management practices (i.e. tree replacement and harvest rotation length extension) and wood product pool were not considered or included.

advantageous in that it integrates the judicious selection of tree species in forestation, the wood harvesting practices, the structural composition of forests, and the dynamics of tree demographics in simulations using a process-based model (i.e. DLEM). We advocate the pressing necessity for the swift implementation of optimal forest management strategies to enhance carbon sequestration in China's forests.

## Methods

### National forest inventory (NFI) data

About 415,000 permanent plots were set up in China for routine survey every five years. Among these plots, over 50,000 surveyed plots were covered with trees during the period of the 6$^{th}$ (1999–2003) to 9$^{th}$ (2014–2018) NFI (Supplementary Fig. S1). Each of these forest plots was 300–600 m$^2$ in size, and the inventory was designed to cover mainland China with a 5-km grid. Information including the land use type, tree species, growth status, forest origin, disturbances, and age were recorded during each NFI. For each plot, trees with a diameter at breast height (DBH) of ≥5 cm were labeled, measured, and recorded. For each tree in each plot, the volume was calculated by referring to the one-variable tree volume tables for each species specifically developed in each province[43]. The stand volumes were summarized to the plot level from all recorded trees. The dataset used in this study is consisted of 18,116,071 tree records

**Table 1 | Forest biomass carbon sink under different management scenarios**

| SSP scenario[a] | Carbon sink of 2020–2100 (Tg C yr$^{-1}$) | Additional sink after implementation of forest practice (Tg C yr$^{-1}$)[b] | | Carbon stock loss due to delayed implementation of both practices (Tg C)[c] | | |
|---|---|---|---|---|---|---|
| | | Tree Rep | Rota Ext | N + 5 | N + 10 | N + 15 |
| SSP1 | 145–159 | 0.23–1.0 | 4.3–4.7 | 47.9–56.6 | 106.7–113.1 | 130.3–140.8 |
| SSP2 | 158–173 | 0.17–1.0 | 4.5–5.0 | 44.3–62.5 | 113.4–131.6 | 136.3–152.6 |
| SSP3 | 172–188 | 0.28–1.1 | 4.8–5.3 | 45.9–66.9 | 116.1–132.4 | 147.0–162.8 |
| SSP5 | 183–200 | 0.17–1.1 | 5.2–5.6 | 43.9–54.7 | 124.1–134.5 | 149.1–166.2 |

[a]SSP1-SSP5 indicate the scenarios of SSP1-2.6, SSP2-4.5, SSP3-7.0, and SSP5-8.5, respectively.
[b]Tree Rep: tree replacement; Rota Ext: harvest rotation length extended by 5 years after key year.
[c]Carbon loss of delayed implementation of management practices was derived from the difference of carbon stock in 2100 between implementation scenarios, N indicate the implementation year and "N + 5", "N + 10", and "N + 15" indicate the postponing implementation of both management practices by 5, 10, and 15 years, respectively.

collected from over 50,000 tree plots surveyed in the 6[th], 7[th], 8[th], and 9[th] NFIs covering a period from 1999 to 2018.

### Forest carbon stock in the 9[th] NFI

Individual tree volume was first converted to individual tree biomass using the DBH-based volume-biomass equations (Supplementary Table S2). The tree biomass was then converted to forest carbon stock for each plot using species-specific conversion ratios (Supplementary Table S3). The relationship between carbon stock and stand age was described using both logarithmic and logistic equations, and the equation with a higher $R^2$ was adopted (Supplementary Figs. S3–S5). Forest carbon stock in 2018 was derived from the 9[th] NFI using the equations and ratios listed above. The species-specific forest carbon stocks in 2018 served as a reference for validation of the simulated results from the process-based modeling (Supplementary Fig. S6).

### Carbon stock/sink derived from statistical model

We established stand-level carbon stock growth models for forests with differed origins (i.e. planted and natural forests), site classes (i.e. 3–9 site classes based on the regional distribution of species and the number of sample plots), and regions (i.e. north or south regions), using three theoretical growth equations (i.e. Richard, Korf, and Hossfeld)[27,44]. These approaches were validated in previous studies[27,44]. First, the NFI plots were classified by DBH with 2 cm interval by species, origins, and provinces. For each DBH class, NFI plots were further divided into 3–9 classes according to the sample number. Second, the three growth models were applied to each species in each region/origin. These models were carefully validated using measured field data. More specifically, for each tree species/species group, the forest plots were divided into ten parts, and model parameterizations were conducted nine times for each of the Richard, Korf, and Hossfeld equations. Each of the ten parts was used as validation samples during parameterization, and the one demonstrating the highest performance from each of the Richard, Korf, and Hossfeld groups was selected for prediction of future carbon stock. Third, the predictions from the three best equations were averaged using $R^2$ as the weighting factors. This approach incorporates crucial factors such as forest age, stand class, and tree densities. Details of the equations can be found in the supplemental Excel file (Supplementary Data 1).

### Climatic and atmospheric chemical condition datasets

The DLEM is a highly-integrated, process-based ecosystem model driven by multiple factors including climate, atmospheric compositions ($CO_2$, nitrogen (N) deposition), land use and cover change (LUCC), and land management practices (harvest, fertilization etc.). In this study, historical climate data were obtained from meteorological stations and published datasets available for the period 1900–2019[29,45], while future climate data were obtained from CMIP6. To force DLEM model, daily climate variables were resampled at 0.5° × 0.5°, including

the maximum/minimum/average air temperature, and the precipitation data. Specifically, four Shared Socioeconomic Pathways (SSP1-2.6, SSP2-4.5, SSP3-7.0, and SSP5-8.5) outputs from Australian Community Climate and Earth System Simulator (ACCESS-CM2) were considered in model simulations. Other atmospheric chemical components, including atmospheric $CO_2$ concentration, and N deposition data, were retrieved from IPCC historical $CO_2$ data and the North American Carbon Program Multi-scale synthesis and Terrestrial Model Intercomparison Project (https://daac.ornl.gov/NACP). The historical N deposition maps from 1996 to 2015 were provided by Jia et al.[46], which were used as the baseline to extrapolate the N deposition of the period 1900–1995. Specifically, the N deposition was set to a fixed level since 2015 because a former study revealed that overall N deposition has been stabilized due to improved agricultural and environmental policies[47].

### Experimental design for DLEM simulations

DLEM simulations were set up to quantify the biomass carbon storage and sink in China's forests. The impacts of forestation, climate change, rising $CO_2$, N deposition, and forest management were considered. To capture the impacts of historical LUCC, we set up the initial simulation year in 1900. The carbon storage and changes during the historical period of 1900–2020 were intensively calibrated and validated before performing the future projection. The historical carbon stock and sink validations can be found in Yu et al.[29], and the validations of a more recent period during the 9[th] NFI can be found in Supplementary Fig. S6. Similar to a previous study, we first obtained the initial condition of each biome in each grid cell (equilibrium state), which is defined as the interannual variation of a 20-year net flux of C, N, and water less than $1\,g\,C\,m^{-2}\,year^{-1}$, $1\,g\,N\,m^{-2}\,year^{-1}$, and $1\,mm\,m^{-2}\,year^{-1}$, respectively[48,49]. To avoid abrupt changes resulting from mode transition, we applied a 10-year spin-up run before the transient run (1900–2100) using initial state information obtained from the equilibrium run.

We designed four groups of experiments using DLEM simulations to delineate the carbon stock changes in China from 2020 to 2100. For all experiments, the simulation of the period from 1900 to 2019 was forced by the same historical forcing data (e.g. climate, LUCC), while the future period from 2020 to 2100 was driven by different forcing data depending on the designed scenarios. In this study, the first group of experiments were designed to be driven by forcing close to realistic conditions. Specifically, we assumed a tree survival rate of planned planted forest at 47% or 85% and that forest expansion and wood harvest follow the plans and regulations of the National Forestry and Grassland Administration with environmental factors (i.e. climate and $CO_2$) varying under different SSPs (Supplementary Table S5). The second group of experiments was designed to account for the impacts of improved forest management practices (i.e. extended forest harvest age, tree species replacement), which were either implemented alone or in combination (Supplementary Table S5). The third group of

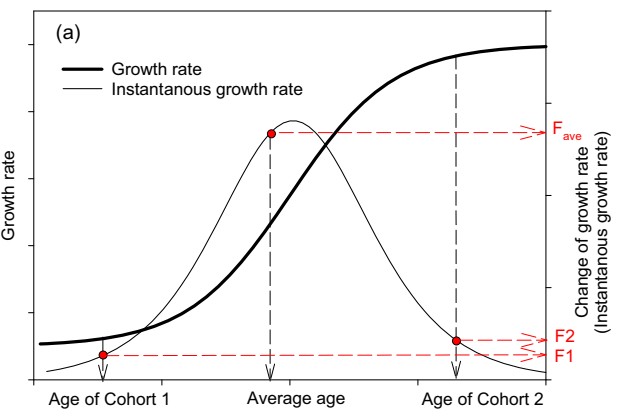

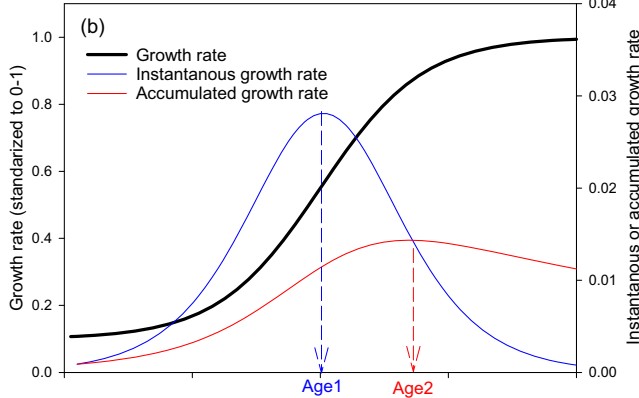

**Fig. 4 | Conceptual diagram of forest age impact on growth.** Panel **a**: changes of growth rate and instantaneous growth rate; panel **b**: relation of the peaks of the instantaneous and accumulated growth rates. F1, F2, and $F_{ave}$ indicate the age factor of cohort 1, cohort 2, and the average age of cohort 1 and cohort 2. Panel **b** adapted from ref. 54, reproduced with permission from SNCSC.

experiments was designed without wood harvest impact, in which the environmental factors were fixed at the 2020 level and no improved forest management practices were implemented (Supplementary Table S5). The fourth group of experiments was designed to keep LUCC fixed at 2020 (i.e. no forest expansion). The group 3 experiments were designed to be compared with the results from statistical model to quantify the impacts of wood harvest on carbon stock/sink. While by comparing the simulated results in groups 1 and 4, the contributions of existing and new forests to carbon stock/sink were quantified. All the simulations were performed at a resolution of 0.5° × 0.5°. The spatial distributions and changes of the forest biomass carbon stock projected from the process-based biogeochemical model are illustrated in Supplementary Figs. S12 and S13.

## Age-cohort dataset for DLEM model

Forest growth was simulated for each species in each of the 0.5 degree grids (~50 km × 50 km). In the original version of the process-based model (i.e. DLEM), the forest age was assumed to be uniform within each grid. Leveraging the NFI dataset, we improved the model by incorporating the diverse range of forest ages within each grid, achieving by considering the effects of tree growth, mortality, harvest, and planting. Specifically, we defined the smallest (basic) unit of each tree species at 1 km², and the tree demographical dynamics were performed at the basic unit and summarized to each 0.5 degree grids. Specifically, the area of each tree species in each grid was divided into multiple age-cohorts, with each unit of area set to 1 km². The demographic information was obtained from the 9th NFI. For example, for *Pinus massoniana*, there were a total of 1878 plots, with age ranging from 1 to 79 years old, recorded in the 9th NFI. We first interpolated to create the *Pinus massoniana* forest age map at a resolution of 1 km × 1 km. The 1878 plots were then allocated to the *Pinus massoniana* forest coverage map using the age map information as reference, during which the number of plots in each 0.5 degree grids was determined by the area in the forest coverage map. This creates forest age data with demographic information matching the 9th NFI, in which the subsequent harvesting, planting, growth, and mortality will be implemented on the age-cohort.

## Age impacts on forest carbon uptake in DLEM model

In previous studies, an adjustment factor was applied to process-based model (i.e. DLEM) to capture age impacts on forest growth[50,51]. However, the parameters were developed for forest group types, and species-specific parameters are not available. Besides, the adjustment

factor was derived from the average stand age and might cause biases (see below explanation). Therefore, in this study, we developed a logistic relationship to describe age impacts on forest growth for each species/species group in each 0.5 degree grid[52,53]. For each of the basic unit of 1 km² age-cohort, the age determines the production capacity of the cohort (Fig. 4a). The total photosynthesis capability of the species/species group in the 0.5-degree grid is the sum of all the age-cohorts. It should be noted that the average impacts of different age cohorts do not equal to the age impacts of the average age of the cohorts [e.g. the average of age factor of cohort 1 (F1) and cohort 2 (F2) vs the age factor of the average age of cohort 1 and 2 ($F_{age}$) showed in the Fig. 4a], signifying the importance of tracking the age dynamics of tree cohorts (i.e. averaged stand age might cause biases in simulations). In this study, when planting and harvesting occurred, the photosynthesis capability of the grids is adjusted based on the demographic information tracked at cohort-level. Generally, there are two key parameters in the logistic equation of tree growth, namely the age that the tree matures and the age that the largest growth rate is attained (Fig. 4b) (refers to Fig. 2 in Salas-Eljatib et al.[54]). As trees grow, forest biomass gradually reaches a relative equilibrium, while the instantaneous growth rate (the first-order derivative) will increase and decrease (blue line in the Fig. 4b). To achieve the largest profit, the wood harvest (Age2 in Fig. 4b) will be performed few years after the peak of the instantaneous growth rate (Age1 in Fig. 4b) because the biomass accumulation can be maintained at a relatively high level even after the peak (red line in the Fig. 4b). According to China's forest harvesting policy, the harvest age is defined as the age that the peak of the average accumulated growth rate (the average of instantaneous growth rate, i.e., Age2 in Fig. 4b) is attained. The harvest age (i.e., the peak of the average accumulated growth rate) is also the intersection point of the average accumulated growth rate curve and the instantaneous growth curve (Fig. 4b). Based on this information, we identify Age1 (i.e. the age of the instantaneous growth peak) using Age2 (i.e. the harvest age, which is also the peak of the average of instantaneous growth rate) for each tree species (Supplementary Table S6). The forest harvesting age (i.e. Age2) were officially released by the State Forestry Administration of China (i.e., Regulations for age-class and age-group division of main tree-species, obtained from: https://www.forestry.gov.cn) (Supplementary Table S6). The age factor on forest growth is defined in the logistic equation:

$$F_{age} = 1 - a \times \frac{1}{1 + e^{\frac{3}{Age1} \times (b - Age1)}} \qquad (1)$$

where $F_{age}$ is the age factor derived from all the cohorts (1 km²) of a species in a 0.5-degree grid, which ranges from (1-$a$) to 1; $a$ is the parameter related to the initial age factor; $b$ is the cohort age; $Age1$ is the peak of the instantaneous growth rate. Since the $F_{age}$ varies between (1-$a$) to 1, a higher/lower $a$ indicate the initial growth rate is lower/higher. The $a$ was initially set to 0.05 assuming the lowest age factor for newly planted/regenerated forest, and further adjusted by referring the carbon sink of process-based model to statistical model in 2020 for each species (Supplementary Fig. S14, Supplementary Table S7).

Note that the daily tree growth was simulated in the process-based model (i.e. DLEM) involving biophysics (e.g. energy and nutrients exchanges within and between ecosystems, such as soil physics, radiative transfer, water and energy flow, momentum movement etc.), physiological (e.g. plant phenology, C and N assimilation, respiration, allocation, and turnover etc.), and biogeochemistry processes (e.g. soil microbe's activities, mineralization/immobilization, nitrification/denitrification, decomposition etc.). The age impacts were applied on the daily growth of tree.

### Forest harvest

China shifted the primary purpose of forest management from timber production to forest sustainability in the late 1990s, and the Commercial Harvest Exclusion policy was further implemented in 2014[55]. Thus, in reality, forest harvest has been and will continue to be limited to a specific area of China's forest, i.e. there are forests free from harvest enforced by policy. Accordingly, we separated each tree species into timber forest and non-timber forest for both natural and planted forests. Harvesting was only performed on mature timber forest in simulations. The harvest age for each tree species was obtained from the State Forestry Administration of China (Supplementary Table S6)[24]. The areas of timber forest for each species were obtained from the China Forest Resource Report 2014–2018[56]. Besides, to ensure that forests were not overexploited, the total harvested forest was limited to the average annual allowable cut during a rotation period. For example, the mature age of the planted *Pinus massoniana* timber forest is 36 years, and we therefore limited the harvested area of the forest to not exceed 1/36 of the total area each year for a certain gridcell. This was done to align the simulation with the real-world situation where not all timbers are harvested immediately upon reaching maturity. This is also in alignment with the regulations formulated by China's government, according to which timber forests are stringently managed to avoid over-harvesting by the National Forestry and Grassland Administration.

### Tree mortality

Carbon loss due to tree die-off has been considered in this study. The national annual tree mortality was at 0.084% derived from the 9th NFI[56]. In model simulations, the mortality rate was randomly applied spatially to trigger tree mortality for each species. Thus, for each tree species in each 0.5-degree grid cell, there is a chance of mortality occurring in a 1-km² stand (e.g. an age cohort), which will trigger biomass carbon loss into the atmosphere in model simulations. After a die-off event, trees will naturally regenerate or be planted for non-timber and timber forests, with species remaining unchanged. As reported that tree mortality might increase due to higher climatic stresses and more frequent climatic extremes (e.g. pests, fires)[57,58], we designed model simulations with annual tree mortalities at 2-, 3-, 4- and 5-time of the rate in the 9th NFI (0.084%) under SSP1-2.6, SSP2-4.5, SSP3-7.0, and SSP5-8.5.

### Forest age increment

Due to lack of tree demographic data, past studies often assumed that forest age increased by 1 each year. For example, a forest plot with an age of 20 years in 2020 will be 60 years old in 2060. This assumption is more applicable to monoculture timber forests such as *Pinus* timber forest, which are routinely harvested with fixed rotation years. However, this might be unsuitable for non-timber forests since the age distribution will depend on factors such as mortality, regeneration, and natural growth (Supplementary Fig. S9). We examined the age changes in 50,000 fixed plots and found that the age increment largely deviated from 1 year per year in mixed non-timber forests (Supplementary Table S8). Thus, in this study, the age increments of mixed non-timber forests were derived from the surveyed plots (Supplementary Table S8), which were used in projections of carbon dynamics using process-based model simulations. Specifically, the annual age increments ($I$) of $N$ mixed non-timber forest plots were assumed to follow a normal distribution centered at "Age accrual per year" ($I_{ave}$): $I \sim N(I_{ave}, \sigma^2)$; where the $I_{ave}$ is the average accrual per year of the species, $\sigma$ is the standard deviation of the $I_{ave}$. Both the $I_{ave}$ and $sd$ were species-specific and derived from the NFI plots to represent the impacts of disturbances on age increment in mixed non-timber forests.

### Wood product residence time

The harvested carbon enters the wood product pool, in which the carbon residence times were product-dependent. The wood products were aggregated into three categories, namely sawn wood, wood-based panels, and paper/paperboard. Based on the quantity of wood products of China from FAOSTAT and the carbon conversion factors and density of wood products (IPCC, 2014), the percentages of carbon in the wood product pools of sawn wood, wood panels, and paper/paperboard during the period of 1961–2020 were calculated. For the period of 1900–1961, the ratio was linearly interpolated based on the trends of 1961–1980. For the period from 2021 to 2100, we assumed that the three wood product ratios remain the same as in 2020. When the wood products reach the end users, the wood products would gradually decay and emit carbon into the atmosphere. The decay of carbon in wood products was calculated by applying the Tier 2 method proposed by IPCC[40]. The first-order decay function used was:

$$C(i+1) = \sum_{j}^{3} C_j(i+1) = \sum_{j}^{3} e^{-k_j} \cdot C_j(i) + \frac{1 - e^{-k_j}}{k_j} \cdot Inflow_j(i) \quad (2)$$

$$k_j = \frac{\ln(2)}{HL_j} \quad (3)$$

where, $i$ = years after the wood product reached the end users;
$j$ = {sawn wood, wood-based panels, paper/paperboard};
$k_j$ = decay constant of first-order decay for category $j$;
$C(i+1)$ = the total carbon stock in wood products in year $i$+1;
$C_j(i)$ = the carbon stock in wood product category $j$ in year $i$;
$Inflow_j(i)$ = the inflow of wood product category $j$ in year $i$;
$HL_j$ = the number of years to lose a half of the wood product category $j$ (35 years for sawn wood, 25 years for wood-based panels, 2 years for paper/paperboard, IPCC, 2006)

### Future forest expansion from official forestation plan

The historical, gridded land-use datasets were developed in our previous study using multiple sources of data, including gridded images from 1887 to 2019, vector maps in the 1980s, and tabular data from 1949 to 2018[28,29]. The database has been corrected for tremendous biases found in existing LUCC data products[29], which serves as a reliable basis for future forestation plan to be implemented. The future forestation was deployed in each province and by year. Specifically, the provincial areas of future plantations were provided by the National Forest Management Plan 2016–2050 released by the State Forestry and Grassland Administration of China (i.e. former National Forestry Administration). According to the forestation plan, new plantations will be established at 15 Mha from 2016 to 2020 and at an additional

49.5 Mha from 2021–2050. The afforestation rate was assumed same as the rate during the 9th NFI in each province. Planted forests might not always survive after planting as forestation failure happens. Therefore, we assumed that tree survival will be positioned between the harsh condition (e.g. arid and semi-arid regions) and the expected criteria designed by the State Forestry Administration. For the period from 2021 onward, two scenarios were considered for forestation, in which the survival rates of the trees were 47% and 85%, respectively. The 47% tree survival rate was derived from the 9th NFI in the arid and semi-arid region of Northwest China, while 85% is the scenario assuming that the survival rate was improved by forest management and land meets the criteria set by the State Forestry Administration[59]. Since China restricts further conversion of croplands and wetland to forestland for food security and biodiversity conservation, we assumed that the future plantations were only allowed to be converted from grassland and shrubland. The newly established forest plantations were divided into timber and non-timber forests according to the timber:nontimber ratio derived from the China Forest Resource Report 2014–2018[57]. Besides, the natural forest area was assumed to increase according to the "Mid- and long-term protection and recovery plan of natural forest in China (2022–2035)" released by the National Forestry and Grassland Administration of China. Specifically, the natural forest will increase from 126.06 Mha in 2020 to 156.2 Mha in 2050 through natural regeneration, conservancy, and protection. See forest distributions in Supplementary Fig. S15.

### Tree species for future forestation

The climate change impacts on the choice of tree species were also less considered in earlier studies. Habitats of tree species will be shaped by climate change, resulting in shifts of land suitability, and the choice of tree species should therefore be identified. Thus, species-specific land suitability should be evaluated before accurate projection of carbon accumulation in planning new plantations can be obtained. Nonetheless, mapping land suitability for tree species is challenging due to limited samples of observational distribution for each species, especially when natural and planted forests need to be separated. In this study, species distribution models (SDMs) were applied to determine the suitability of each tree species in each gridcell. Specifically, the state-of-art ensemble machine learner–random forests (RFs) were implemented to delineate forest distribution for the baseline period of 2010–2020 (baseline) and to project future (2090–2100) distributions under an ensemble forecasting framework. Similar to previous studies[35], the classification tree algorithm was used to develop RF models using the R package "randomForest" by linking forest distribution with climate data. The tree distribution information from the NFIs were used to train the SDMs, and future climate data of the four Shared Socioeconomic Pathways (SSP1-2.6, SSP2-4.5, SSP3-7.0, and SSP5-8.5) derived from CMIP6 were adopted to project the land suitability index for each tree species at a 1 km × 1 km resolution. Examples of the habitat distributions of *Pinus massoniana, Pinus sylvestris*, and *Picea spp.* were provided in Supplementary Fig. S16. The grid-cell with higher suitability index for a specific species was given priority to be planted with the tree species. This approach ensures that the appropriate tree species were selected for future plantations by considering climate change impacts. Specifically, the number of samples used in developing the land suitability index for each species is listed in Supplementary Table S4.

### Forest management practices for carbon stock enhancement

Two major management practices were considered for carbon sequestration enhancement. The first practice is to postpone harvesting time to help retain carbon in ecosystem in timber forests. Since carbon removed persistently from timber forests, gridcells with intensive carbon removals could be net carbon sources. Therefore, we identified the year of carbon source for each tree species and reduced the harvesting intensity by postponing the harvesting time by 5 years. The year that the harvesting extension is implemented is hereby defined as the key year, which indicates the year of plummeted carbon sink if free from the practice. Thus, we designed simulations by extending the wood harvesting age by 5 years after the key year for timber forests. The second practice is to replace inappropriate tree species by indigenous species. We examined the species in each forest management zone (see Supplementary Fig. S17) and located the species that are alien and inappropriate according to the forest management guidance released by the State Forestry Administration[15]. A total of 4.13 Mha of forests were spotted to be replaced in 2025. Simulation experiments were designed to replace the inappropriate species with indigenous species with similar habitat requirements (Supplementary Table S9). Specifically, the species replacements are listed in Supplementary Table S9. Based on these criteria, the spatial distribution of the implementation of tree replacement practice is illustrated in Supplementary Fig. S18, in which the value indicated the forest percentage to be replaced.

### LUCC data and model validation

The historical LUCC dataset has been intensively validated in former studies both spatially and temporally[28,29]. This study further partitioned forests into different species based on the presence of harvest (i.e. timber forest and non-timber forest). For biomass and soil carbon stock simulations during the historical period (1900–2019), rigorous calibration and validation have also been conducted using measurement data collected from a nationwide field campaign in China[29]. In this study, we further compared the estimates of species-level carbon stock from process-based model and the statistical model using the 9th NFI data (Supplementary Fig. S6).

## Data availability

The projected forest coverage data generated in this study is available from https://doi.org/10.6084/m9.figshare.25323175.v1. Grassland and shrubland maps used were the China Land Use and Cover Change dataset, which can be obtained from the Data Center for Resources and Environmental Sciences, Chinese Academy of Sciences: http://www.resdc.cn. The climate data is available from https://cds.climate.copernicus.eu. The atmospheric $CO_2$ concentration and nitrogen deposition data were retrieved from IPCC historical $CO_2$ data and the North American Carbon Program Multi-scale synthesis and Terrestrial Model Intercomparison Project at: https://daac.ornl.gov/NACP. The nitrogen deposition during 1996–2015 were updated using data provided at: http://www.nesdc.org.cn/sdo/detail?id=5fa53685042ebb70d0c83403.

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

## Acknowledgements

The authors thank the National Forestry Administration for providing the forest inventory data. The authors would like to extend their gratitude to the more than 20,000 surveyors who participated in the field data collections. This study was supported by National Key Research and Development Program of China (No. 2021YFD2200405 (S.R.L.)), and China National Science Foundation (No. 32361143869 (Z.Y.), 32371663 (Z.Y.), and 32001166 (Z.Y.)).

## Author contributions

Z.Y. and S.R.L. conceived and designed the research. Z.Y. performed process-based model simulations and drew the figures. H.K.L. and Y.L.D. performed simulation using statistical model. W.G.L. calculated carbon residence in wood products. Z.Y. and S.R.L. wrote the first draft together. J.J.L. and E.A. extensively edited the manuscript and provided suggestions to revise the graphs. S.L.P., J.J.L., P.S.S., C.Q.L., S.S., W.B.Y., G.Y.Z., and H.Q.T. provided essential suggestions to improve the manuscript. All co-authors reviewed and contributed to the revised manuscript.

## Competing interests

The authors declare no competing interests.

## Ethics approval

All collaborators of this study have fulfilled the criteria for authorship required by Nature Portfolio journals have been included as authors, as their participation was essential for the design and implementation of the study. This research does not result in stigmatization, incrimination, discrimination or personal risk to participants.
