## [Peer Review File · Nature Communications]

Maximizing carbon sequestration potential in Chinese forests through optimal managementREVIEWER COMMENTS

Reviewer #1 (Remarks to the Author):

This study evaluated the forest biomass carbon sequestration of China in 2020-2100 by using forest inventory data and process model. It found that the biomass carbon stock will increase from 14.0 ± 1.8 to 24.5 ± 1.6 Pg and the peak of sequestration time will be delayed by expanding forest area, regulating timber harvest and replacing with suitable tree species. It concluded that it is imperative to expedite the implementation of C sink enhancement practices to maximize C sequestration potential in Chinese forests.

The research topic is important for the best practices of forest management in China towards the nation's goal of carbon neutrality and forest sustainability.

The study has done a substantial amount of work including data syntheses and modeling experiments. Many factors are considered and many different scenarios are used, which should be the strength of the paper that "collectively addressed the vital components in an integrated manner". However, this complexity made the manuscript very difficult to follow. I had a hard time to understand the results and discussion part. I had to look for the information between here and the method section back and forth, back and forth, to understand which results are corresponding to which model/parameter settings. Overall, the writing structure and methods need great clarification.

In the manuscript, it is not clear if they solely used DLEM model (process-based model means DLEM)? Or they also used InTEC model? And based on what modeling experiments the final findings are presented?

In the results of figures 1, 2, 3, what modeling scenarios were used to get these results? In Table 1, there are four SSP scenarios. What about the other parameters, tree species, harvest, climate, CO₂...? The same thing, the main conclusion " 172.3 ± 16.9 Tg C yr⁻¹" and " 28.1 ± 0.4 Tg C yr⁻¹" sequestration rate are based on what kind of settings of the modeling? What are the parameters(values) for optimal management? Optimal management is not defined.

Line 38, an expansion of biomass carbon stock was found, "surpassing the average values in previous studies...", and Line 42, "our findings bring to light an overlooked potential...". These two sentences read contradictory.

Line 54, inconsistent, the inconsistent findings due to a lack of optimal management plan? should be different setting or consideration of plan but not lack of plan?

Line 69, which year of spatial data was used as initials for modeling? Modeling work is not mentioned in

the last paragraph and the whole introduction. What is the “optimal management”, should define.

Results and Discussion

This section is hard to understand. The results and points of discussion are very scattered.

Line 73: what is the aim to optimize timber harvest?

Line 74: reduced large uncertainties of what?

Line 76: do you have those maps? Give an example of a species map would be good.

Line 86: do you have the maps for existing and new forests?

Line 98: habitat suitability maps, in what grid size the species habitat suitability maps are generated?

Line 105: what two methods?

Line 108: what are all scenarios?

Line 115: DLEM model or InTEC model, or both?

Line 142, what are short-, mid- and long-lived wood products?

Figure 1, process-based model is which model?

Line 162-164, Figure 2, how much carbon sink is from exiting forest and new forest, respectively?

Line 285, what is the significance to compare with Canada? Why not other countries?

Line 289-291: this may be a weak point for the conclusion.

Line 422-423, the mortality rate was randomly applied. How many times of random sampling/modeling are done?

Line 532: what is the size of the grid-cell? Are the species mixed growing in one grid or they are treated as monoculture in one grid? 0.5 degree? 5 km?

Reviewer #2 (Remarks to the Author):

Review of “Maximizing carbon sequestration potential of Chinese forests through optimal management.”

General Comments: This is a well-written, thoughtful analysis of the ability for Chinese forests to store carbon. The authors did an excellent job of taking into account all of the confounding (and explanatory factors) for how the future forest carbon sink can behave. I only have a few recommendations and I am strongly supportive of the publication of the manuscript.

Abstract: The increase is noteworthy, but how is it occurring? Is this just regrowth? Or enhanced growth? Or harvest decreases? It would be useful in the abstract to have a primary reason for the increase.

Text: Can you indicate somewhere in the main text what process-based model was used? I know it is in the methods, but it would be nice to have it mentioned in the text with a brief description of what processes are included. This is important given the ability of the model to include enhanced growth under higher co2 conditions, etc.

The sink estimate is quite a bit higher than previous estimates, but there are many reasons as stated by the authors. In the paragraph where the sink strength is compared to the US and the EU, can you put the estimates in a per unit area basis for direct comparison? I think this would help with perspective (e.g., is the forest area in China X times bigger than the US or EU??)

Great study!

Reviewer #3 (Remarks to the Author):

In general, Supplementary materials were very rich but the methods were too often elusive : how did the author validate the accuracy of the age correction factor? What the input data used for the process-based model and where do they come from? In the main text (introduction, results and discussion) there is often a confusion between recent trend and future projections and this is the task of the reader to understand the period investigated. Most importantly, I do not believe that the authors actually reduced the uncertainties of the C sequestration of Chinese forest. Indeed the uncertainty analysis actually corresponds to a sensitivity analysis of the effect of various scenarios on forest C sequestration. To really speak of reducing uncertainties, more space and time should be dedicated to the validation of the model performance. This can only be done by comparing model simulations against independent observation data. This would imply to split the dataset in two subsets: one for model calibration, the other for model validation. This would be a key result that should be presented in the core of the manuscript.

Hereafter I provide some detailed remarks following the text:

Abstract

General remark : Please, better define the distinction between potential C sequestration in forest biomass and what it would take to reach this potential in terms of forest management practices under various RCP scenarios.

Line 36-37 : « Our projections indicate a remarkable increase in biomass carbon stock », under which conditions/assumptions ?

Line 37-38 : same remarks, what would it take to reach this biomass C stocks in 2100 (no harvest ? sustainable harvest ? under which RCP scenario ?)

Introduction

Lines 53-62 : Here again, it is a bit confusing to mention the uncertainties regarding the estimates of C stocks and sink in forests together with the lack of « optimal forest management plan and strategies ». Please differentiate the research-gap associated to the monitoring of Forest C stocks and sinks (recent trend) and the research-gap regarding the assessment of the potential C sequestration in forests. These

are two different problems which should be clearly distinguished even though they are linked to one another.

It is also not clear if the author are focusing on forest biomass C stocks or forest C stocks (including biomass, dead wood, litter, soil). Please specify.

Line65-70 : So the time-frame of the study is 1999-2018 ? I do not understand what are the objectives of the study. Is it to estimate the forest C sink in forest (biomass ?) over the 1999-2018 period in China ? Please present more clearly : (i) the research-gap(s), (ii) the research question(s), (iii) the research objective(s). In its current form the introduction is too elusive and does not enable to understand the purpose of the research.

A brief presentation of the method should be presented at the end of the introduction to understand what has been performed. This would be the end of the introduction (iv) how these objectives were reached.

Results and Discussion

Lines 73-74 : Unclear what has been performed. If wood harvest has been derived from forest growth and regrowth, this should be presented as a result and compare with independant data on forest wood harvest, either from national statistics or at least from the FAO database (at the national level).

Lines 75-76 : Suddently we move from recent forest C sink to future forestation. Again, the data frame is very obscure. What is the temporal projection ? 2100 ?

Lines77-78 : What are the studies using « initial carbon stock validated by inventory data » that you are mentioning here ?

Line 84-86 : Which process-based model did you use ?

Lines 86-89 : What are the tree survival rate scenrios ? How were they established ? What kind of statistical model did you use ? Which trajectories of wood harvest, CO2 conc and climate change did you implement ? How did you choose them ? Even tough the methods are at the end of the paper, what you have performed should be clearly stated before each results.

Line 89 : What are those « vast majority of other studies » that you are refering to ?

Lines 104-106 : Will increase to 21.6-24.3 PC in 2100 under which assumentions of climate change, afforestation, wood management ? It is also very obscure how did you reduce the range in uncertainties compared to other studies. The approach implement to reduce the uncertainty range should be stated very cleary. It is really unclear how you validated your models, knowing that model simulations can ONLY be validated against independant observation time-series data.

Lines 113-129 : This whole passage remains completly obscur aslong as we don't know which models were used, which are the controlling (inputs) variables of the models, following which assumptions were the value of these input data set, and, most importantly, how the uncertainties of model input AND

output data were estimated. Again these uncertainties can, by no way, be taken as the envelope of the model results as long as the models have not been first evaluated against observational data.

Line 127-128 : No timber harvest does not promote forest C sequestration, it only rejuvenate forests so that there is obviously a C sink due to forest regrowth. But this forest regrowth can be much less efficient than letting forests age as much as possible. Because even though the C sink may diminish with forest ageing, the important is not the C sink but the C stocks which transit between forest ecosystems – timber goods – the atmosphere !

Lines 140-141 : which categories of wood products have been distinguished ?

Figure 1. I do not understand how there can be « inventory-validated studies » for 2030-2060 (cyan box panel a) ? So the range of uncertainties reflect the divergence between various scenarios of tree survival rate and other « different scenrios » for the process-based model ? This is really not an uncertainty estimates but a sensitivity analysis to different choice. It is an overstatement to say that the present research enable to reduce uncertainties regarding forest C sink in China.

Lines 159-167 : Please provide a clear definition of non-timber natural forest », « timber natural forest », « non-timber planted forest », and « timber planted forest ».

Figure 2. It is a bit surprising to see that there is strong spatial overlap of the different forest categories... Could you explain this in the legend.

Lines 174-176 : Please stay with Pg and not change unit.

Line 184 : Forest areas in previous reports were either much smaller than what they are in reality. But we do not know what the rality of forest area will be in 2060. This simply means that you had a more optimistic assumption regarding forest area expansion in this study.

Line 186 : Historical forest maps were rigorously calibrated and validated. This should be presented before since model validation is also a result. Therefore, it should come in the result section and a brief explanation of how this was performed need to be presented even though the details are further developped in the methods.

Line 197 : If timber forest are pemanently rejuvenating, the C sink is also permanently compensated by a C source, so that the C sink is not persistent.

Line 210-212 : This statement is largely incorrect, except if timber harvest is used for very long-term wood products. Please see Erb, H., Haberl, H., Le Noë, J., Tappeiner, U., Tasser, E., Gingrich, S., (2022). Changes in perspective needed to forge “no-regret” forest-based climate change mitigation strategies. Glob. Change Biol. Bioenergy. 12921. DOI: 10.1111/gcbb.12921

Figure 3. First time that the SSP are mentioned... Were they implemented in the previous figures shown?

Line 268: 43.9-166.8 TgC accumulated from 202-2100, this large range of values is in contradiction with the results shown in Fig. 1a?

Table 1. Please use the same unit everywhere, this is very confusing otherwise. Are you referring to SSP or RCP? As far as I understand it, SSP are socio-economic pathway while RCP are scenarios of C emissions. Of course, both can be coupled but it seems more adequate to refer to RCP rather than SSP in this case.

Method

Lines 324-325: What are those equations. On which variables do they rely and how did you get access to these variables? This is really too elusive and we can't understand what has been performed here

Line 328-331: Which data were used to validate the models? Saying that careful validation was performed is really not enough and not convincing at all.

Lines 333-337: But which process-based model did you use? What are the equations of the model? What input variable are required to run it?

Lines 351-390: It is difficult to understand which data were used to implement the age correction factor and how this could be species-specific. If I understand well authors have optimized the a and b parameters based on the C sink simulated by the process-based and statistical mode. But this sounds very tautological because the a and b parameters should be implemented within the process-based model. How is this validated against observation data?

Lines 393-399: Which data have been used to implement the biophysics, physiological and biogeochemistry processes: what are the input data and main parameters?

Lines 420-430: Tree mortality may also depend on forest age but ok.

Lines 481: 15Mha from 2016 to 2020: has this really occurred?

Lines 505-508: to me this sounds more like RCP rather than SSP scenarios. Maybe correct

Lines 561-562: But we do not understand how the calibration and validation have been performed

Supplementary materials

What is the source of the parameters presented in Tables S2-3?

Response to reviewer comments

We thank reviewers for their precious and constructive suggestions to improve this manuscript. We have addressed all the comments raised by the three reviewers. Please find our point-by-point response below.

Please note that the line numbers refer to the revised manuscript uploaded into the system. Also, while the reference/citation numbers have been updated and corrected, these changes are not shown in the manuscript file to assist the review of the revised version because the reference management software changes all references/citations even for just one change.

REVIEWER COMMENTS

Reviewer #1 (Remarks to the Author):

This study evaluated the forest biomass carbon sequestration of China in 2020-2100 by using forest inventory data and process model. It found that the biomass carbon stock will increase from 14.0 ± 1.8 to 24.5 ± 1.6 Pg and the peak of sequestration time will be delayed by expanding forest area, regulating timber harvest and replacing with suitable tree species. It concluded that it is imperative to expedite the implementation of C sink enhancement practices to maximize C sequestration potential in Chinese forests.

The research topic is important for the best practices of forest management in China towards the nation's goal of carbon neutrality and forest sustainability.

Response: We thank the reviewer for valuing our work. We considered all the suggestions, which were very helpful for improvement of the manuscript!

The study has done a substantial amount of work including data syntheses and modeling experiments. Many factors are considered and many different scenarios are used, which should be the strength of the paper that “collectively addressed the vital components in an integrated

manner”. However, this complexity made the manuscript very difficult to follow. I had a hard time to understand the results and discussion part. I had to look for the information between here and the method section back and forth, back and forth, to understand which results are corresponding to which model/parameter settings. Overall, the writing structure and methods need great clarification.

Response: We thank the reviewer for the constructive comments. Indeed, we have considered many factors and many different scenarios to achieve the goal of “collectively addressed the vital components in an integrated manner”. Therefore, unavoidably the information provided was overwhelming. We apologize and have intensively revised the text to improve the readability and clarity. Please check our revisions in the main text.

In the manuscript, it is not clear if they solely used DLEM model (process-based model means DLEM)? Or they also used InTEC model? And based on what modeling experiments the final findings are presented?

Response: We thank the reviewer for the suggestion. We only used DLEM model in the simulations, and we rephrased the Lines 476-477 to avoid the confusion. We used both statistical and process-based biogeochemical models. However, we found that the carbon stock changes derived from statistical model were not realistic since wood harvest and forest age changes were either ignored or not correctly represented in this method. Therefore, the final findings were presented using simulations from process-based model (i.e. DLEM). In DLEM simulations, we designed multiple experiments which can be divided into four groups:

- 1) Group 1: designed to quantify the carbon stock and sink changes without optimized management practices;
- 2) Group 2: designed to quantify the additional carbon stock and sink if forest management is optimized.
- 3) Group 3: designed to examine the bias if wood harvest is neglected.
- 4) Group 4: designed to quantify the sink from existing and new forests.

Therefore, the results presented were from the first group of simulations unless specifically indicated using terms such as “optimized practices”, “improve management”, “additional

carbon sink". To make this clear to reader, we have made revisions throughout the text. Please see our revisions in Lines 35-39, 80-90, 138-140, 151-152, 157, 161-162.

In the results of figures 1, 2, 3, what modeling scenarios were used to get these results? In Table 1, there are four SSP scenarios. What about the other parameters, tree species, harvest, climate, CO₂...? The same thing, the main conclusion "172.3 ± 16.9 Tg C yr⁻¹" and "28.1 ± 0.4 Tg C yr⁻¹" sequestration rate are based on what kind of settings of the modeling? What are the parameters(values) for optimal management? Optimal management is not defined.

Response: We thank the reviewer for the constructive suggestions. As explained in previous response, the results in Figure 1-3 were derived from the first group of experiments, while the results of Table 1 were derived from the second group of experiments. Here are the details:

In Figure 1, the statistical models were used to estimate the carbon stock and sink under scenarios of the two tree survival rates (i.e. 47% and 85% tree survival rates) without climate change. The process-based simulations using DLEM were conducted assuming two tree survival rates (i.e. 47% and 85% tree survival rates) and four SSP scenarios (and four corresponding mortality rates and CO₂ concentrations). In Figure 1, the improved management practices (i.e. tree replacement and harvest rotation length extension) and wood product pool were not considered or included. We have added information in Lines 198-200 for clarifications. The Figures 2&3 were derived from process-based simulations same as in Figure 1. We also added the information in Lines 225-229 and 281-283. Other factors, including tree species, wood harvest, climate, CO₂ were considered in all simulations (i.e. considered in both first and second groups of simulations, see below).

In Table 1, the results were derived from experiments assuming that the management practices were absent, implemented separately, or postponed under four SSPs. A total of 56 simulations were conducted in Table 1 (7 management scenarios × 2 tree survival rate scenarios × 4 SSPs). Regarding other parameters, i.e. tree species, harvest, climate, CO₂ as mentioned by the reviewer, they were considered in all simulations in Table 1. That means all other factors were controlled close to reality. For example, tree species adopted for forestation were based

on habitat suitability maps and the dominant species appropriate for each province. Thus, for a specific gridcell, the forestation species were not changed between all 48 simulations in Table 1. Similar, wood harvests were considered in all simulations but the harvesting cycles were designed differently to quantify the carbon stock gain from improved management. Besides, climate and CO₂ were considered simultaneously in each SSPs because they are interactive, and it will be meaningless to separate these two factors. For example, in SSP1-2.6, both the climate and CO₂ were derived from SSP1-2.6 since the climate change was closely related to the changes in atmospheric CO₂ concentration.

The optimal management indicates the practice ensemble that help achieve the goal of carbon stock and sink enhancement. We also added the explanations about “optimal management” in the last paragraph of introduction. Please check our revisions in Lines 87-90.

Line 38, an expansion of biomass carbon stock was found, “surpassing the average values in previous studies...”, and Line 42, “our findings bring to light an overlooked potential...”. These two sentences read contradictory.

Response: We have deleted the sentence to condense the Abstract. Please check Lines 45-47.

Line 54, inconsistent, the inconsistent findings due to a lack of optimal management plan? should be different setting or consideration of plan but not lack of plan?

Response: We rephrased the sentence. Please check our revision in Line 60.

Line 69, which year of spatial data was used as initials for modeling? Modeling work is not mentioned in the last paragraph and the whole introduction. What is the “optimal management”, should define.

Response: We thank the reviewer for pointing this out. The latest National Forest Inventory (NFI) data in 2018 was used as initial for modeling, and we added this information in Line 79. We added modeling designs and also defined “optimal management” in the introduction for clarity. Please check our revisions in Lines 80-90.

Results and Discussion

This section is hard to understand. The results and points of discussion are very scattered.

Response: We thank the reviewer for the comment. The logic of this section was to elaborate the temporal changes of carbon stock and sink first, and then depict the spatial distribution of the projected carbon dynamics, while the last part was about the carbon stock and sink enhancement from improved management practices. We have intensively revised this section by adding more background and methodological information to help increase the readability. Please check the revised text in this section.

Line 73: what is the aim to optimize timber harvest?

Response: Timber harvest was generally represented as carbon removed from gridcells in the unit of fraction (% of total carbon) or unit of kg C per year in simulations using process-based model. For example, the LUH2 provides wood harvest dataset that has been widely used in different studies (https://luh.umd.edu/LUH2/LUH2_v2f_README_v6.pdf). In the dataset, future wood harvest data were standardized into a total national wood harvest demand in megagrams of carbon per year (Mg C yr^{-1}), as was the fuelwood component of that national wood harvest, either by aggregating gridded wood harvest data into national totals or by disaggregating regional wood harvest data using the ratio of national to regional wood harvest from the end of the historical period (i.e., 2015) (Hurtt et al. 2020). In other words, the wood harvest was demand-driven and using the latest year as baseline for spatial disaggregation. However, since wood harvest is species-specific, in which the harvesting cycles (i.e. rotation length) will be differed by locations and further perplexed by forestation, tree species replacement etc. Therefore, in our simulations, we optimized and determined the timber harvest by species and age, and the harvesting impacts on forest age were designed to further affect the forest demographic and growth. Such optimization improves the carbon removal and accumulation processes in simulations.

Besides, a substantial portion of forests were banned from harvesting, because they were established in ecological fragile regions for environment protection, such as sand fixation,

soil erosion prevention, and biodiversity conservation. We optimized timber harvest by limiting wood removal in timber forests only, which is more realistic.

Reference:

Hurttt, G. C., Chini, L., Sahajpal, R., Froking, S., Bodirsky, B. L., Calvin, K., ... & Zhang, X. (2020). Harmonization of global land use change and management for the period 850–2100 (LUH2) for CMIP6. *Geoscientific Model Development*, 13(11), 5425-5464.

Line 74: reduced large uncertainties of what?

Response: According to former studies (Pugh et al 2018; Cool-Patton et al. 2020), forest regrowth is a large uncertainty source of land carbon sink. For example, Cool-Patton et al. (2020) reported that carbon accumulation rates in regrowing natural forests shows over 100-fold variation across the globe, and the default rates from the Intergovernmental Panel on Climate Change (IPCC) may underestimate aboveground carbon accumulation rates by 32% on average. Moreover, Pugh et al (2018) revealed that it is not possible to understand the current global carbon sink without accounting for the sizeable sink due to forest demography. Both the studies confirmed that a large portion of the current terrestrial carbon sink is strictly transient in nature (i.e. forest regrowth after disturbance). Therefore, our study was purposely designed to capture forest demography, harvest, and species dynamics to reduce the uncertainties resulting from forest growth and regrowth. We have added information in Lines 94-100 to clarify this.

References:

Cook-Patton, S. C. *et al.* Mapping carbon accumulation potential from global natural forest regrowth. *Nature* **585**, (2020).

Pugh, T. A. M. *et al.* Role of forest regrowth in global carbon sink dynamics. *Proc. Natl. Acad. Sci. U. S. A.* **116**, 4382–4387 (2019).

Line 76: do you have those maps? Give an example of a species map would be good.

Response: We thank the reviewer for the suggestion. Yes, we do have the maps, and we provide examples of the three typical tree species widely used in forestation in the supplement. Please see our revision in Figure S16.

Line 86: do you have the maps for existing and new forests?

Response: We thank the reviewer for the suggestion. Yes, we do have the maps of existing and new forests. We added this information in supplement. Please check Figure S15.

Line 98: habitat suitability maps, in what grid size the species habitat suitability maps are generated?

Response: The habitat suitability maps were generated at 1 km² originally. We added this information in Lines 656-657.

Line 105: what two methods?

Response: The two methods are the statistical and process-based models. We added this information in Line 137.

Line 108: what are all scenarios?

Response: "All scenarios" are experiments with different tree survival rates and SSPs. We rephrased the sentence in Line 145 to make this clear.

Line 115: DLEM model or InTEC model, or both?

Response: Simulations from DLEM model only. Text revised in Line 151.

Line 142, what are short-, mid- and long-lived wood products?

Response: The short-, mid- and long-lived wood products are paper/paperboard, wood-based panels, and sawn wood, respectively. We have added this information in Lines 182-183.

Figure 1, process-based model is which model?

Response: The process-based model is DLEM, and we added this information in Line 198 for clarity.

Line 162-164, Figure 2, how much carbon sink is from existing forest and new forest, respectively?

Response: We thank the reviewer for the suggestion. We set up more experiments and quantify that the sink from existing and new forests are $72 \pm 2\%$ and $28 \pm 2\%$, respectively. To do this, we designed experiments to keep land use and cover change fixed at 2020. By comparing the new simulations with experiments in group 1, we identified the contributions of the existing and new forests. We added the results in Lines 215-217.

Line 285, what is the significance to compare with Canada? Why not other countries?

Response: We thank reviewer for capturing this. We compared the forest carbon sink in China and other regions such as Europe, U.S., and Canada in the Results and Discussion section. This is because these four countries/regions are in similar latitude ranges with similar forest types and relatively close spatial areal coverage. We did not compare our results to countries, like for example, Brazil and Australia, because the forest types, climate, and coverages are very different.

Line 289-291: this may be a weak point for the conclusion.

Response: We added a summary at the end of the section. Please check our revisions in Lines 355-361.

Line 422-423, the mortality rate was randomly applied. How many times of random sampling/modeling are done?

Response: In the simulations forcing by each of four SSPs, i.e. SSP1-2.6, SSP2-4.5, SSP3-7.0, and SSP5-8.5, the mortality rate was randomly applied spatially in each gridcell at 2-, 3-, 4- and 5-time of the national annual tree mortality in the 9th NFI (0.084%). Therefore, a total of 64

experiments were conducted, including 8 experiments (2 survival rate × 4 SSPs) in group 1 and 56 experiments (7 management scenarios × 2 tree survival rate scenarios × 4 SSPs) in group 2.

Line 532: what is the size of the grid-cell? Are the species mixed growing in one grid or they are treated as monoculture in one grid? 0.5 degree? 5 km?

Response: The size of the grid-cell in model simulations was 0.5 degree (approximately 50 km × 50 km). Each tree species was treated as a separate type consisting of different age cohorts in one grid. For example, *Pinus massoniana* was treated as a monoculture, while Broadleaf mixed forest was a mixed type.

Reviewer #2 (Remarks to the Author):

Review of “Maximizing carbon sequestration potential of Chinese forests through optimal management.”

General Comments: This is a well-written, thoughtful analysis of the ability for Chinese forests to store carbon. The authors did an excellent job of taking into account all of the confounding (and explanatory factors) for how the future forest carbon sink can behave. I only have a few recommendations and I am strongly supportive of the publication of the manuscript.

Response: We thank the reviewer for the constructive comments and valuing our work.

Abstract: The increase is noteworthy, but how is it occurring? Is this just regrowth? Or enhanced growth? Or harvest decreases? It would be useful in the abstract to have a primary reason for the increase.

Response: We thank the reviewer for the suggestion. The harvest intensity is relatively consistent in the projection. The increase is contributed from natural and regrowth (85%) and the enhanced growth from warming (15%), with natural growth being the primary contributor. Since the abstract exceeds the word limit, we added the reason of the increase in the main text (i.e. Lines 140-142).

Text: Can you indicate somewhere in the main text what process-based model was used? I know it is in the methods, but it would be nice to have it mentioned in the text with a brief description of what processes are included. This is important given the ability of the model to include enhanced growth under higher co2 conditions, etc.

Response: We thank the reviewer for the suggestion. We added the information in the main text. Please check our revisions in Lines 114, 139, 151, 156, 205.

The sink estimate is quite a bit higher than previous estimates, but there are many reasons as stated by the authors. In the paragraph where the sink strength is compared to the US and the EU, can you put the estimates in a per unit area basis for direct comparison? I think this would help with perspective (e.g., is the forest area in China X times bigger than the US or EU??)

Response: We thank the reviewer for the suggestion. According to FAO (<https://fra-data.fao.org/assessments/fra/2020/EU/home/overview>) and the national statistics, the forest areas of the U.S., EU, and China were 310, 1017.5, and 220 Mha in the year around 2020. However, the EU's forest sink cited was the future sink, but the forest area was not reported in the study. To avoid misleading information, we only compared the U.S. and China. Therefore, we calculated per unit area basis of carbon sink based on forest area data of the U.S. and China. We added this information in the text as suggested. Please check our revisions in Lines 235-236.

Great study!

Response: We really appreciate the reviewer for valuing our work.

Reviewer #3 (Remarks to the Author):

In general, Supplementary materials were very rich but the methods were too often elusive : how did the author validate the accuracy of the age correction factor? What the input data used for the process-based model and where do they come from? In the main text (introduction, results and discussion) there is often a confusion between recent trend and future projections and this is the task of the reader to understand the period investigated. Most importantly, I do not believe that the authors actually reduced the uncertainties of the C

sequestration of chinese forest. Indeed the uncertainty analysis actually corresponds to a sensitivity analysis of the effect of various scenarios on forest C sequestration. To really speak of reducing uncertainties, more space and time should be dedicated to the validation of the model performance. This can only be done by comparing model simulations against independant observation data. This would imply to split the dataset in two subsets: one for model calibration, the other for model validation. This would be a key result that should be presented in the core of the manuscript.

Response: We thank the reviewer for the constructive comments. We considered them all and improved the manuscript accordingly. Here are the responses to the reviewer's overall concerns:

1) The age correction factor:

We did not validate the parameters because the age factors were derived from observational data. The parameters were developed based on forest quantitative theory. As we know, tree growth follows the logistic equation, which can be generally divided into three phrases (see Figure 1a below, similar to Figure 2 in Salas-Eljatib et al. (2021)). Trees grow relatively slowly at first, increasing their growth-rate to the point of inflection of the growth curve. In phase 2, the growth is fast, and in phase 3, there is a decrease of the growth-rates.

There are two increment rates. The first one is the instantaneous growth rate (i.e. current annual increment), which is the difference in the growth at the beginning and the end of the year (blue curve in Figure 1b below). The second is the accumulated growth rate (i.e. mean annual increment), which is found by dividing the growth through time t by the number of years required to produce it (red curve in Figure 1b below).

To achieve the largest profit, the wood harvest is not conducted at the year of *Age1* (i.e. the year of highest instantaneous growth rate), but it will be performed few years after at *Age2* (i.e. the year of highest accumulated growth rate) because the biomass accumulation can be maintained at a relatively high level even after the peak (red line in Figure 1b below). Therefore, according to China's forest harvesting policy, the harvest age is defined as the age that the peak of the average accumulated growth

rate (*Age2* in Figure 1b below) is attained. The forest harvesting ages were officially released by the State Forestry Administration of China (i.e., Regulations for age-class and age-group division of main tree-species, obtained from: <https://www.forestry.gov.cn>) The harvest ages for each species/species group were determined by decades of experiences and observational data obtained from field. Therefore, the parameters *a* and *b* were derived from official document that has solid basis.

Figure 1. Growth curve and the changes in tree growth rate (Adapted from: Growth equations in forest research: mathematical basis and model similarities, Salas-Eljatib, C., Mehtätalo, L., Gregoire, T. G., Soto, D. P., & Vargas-Gaete, R., Current Forestry Reports, 7, Springer Nature, 2021, reproduced with permission from SNCSC.)

2) Input data for process-based model:

The major input data required for the process-based model (DLEM) include:

- a) climate data: the maximum/minimum/average air temperature, solar radiation, and precipitation data. These climate variables were resampled to $0.5^{\circ} \times 0.5^{\circ}$. Four Shared Socioeconomic Pathways (SSP1-2.6, SSP2-4.5, SSP3-7.0, and SSP5-8.5) outputs from Australian Community Climate and Earth System Simulator (ACCESS-CM2) were obtained from: <https://cds.climate.copernicus.eu/cdsapp#!/dataset/projections-cmip6?tab=overview>.

- b) atmospheric chemical condition data: atmospheric CO₂ concentration, and nitrogen deposition data. retrieved from IPCC historical CO₂ data and the NACP MsTMIP (<https://daac.ornl.gov/NACP>). The historical N deposition maps from 1996 to 2015 were provided by Jia et al (2018), which were used as the baseline to extrapolate the N deposition of the period 1900 to 1995. Specifically, the N deposition was set to a fixed level since 2015 because a former study revealed that overall N deposition has been stabilized due to improved agricultural and environmental policies (Yu et al 2019).
- c) Land use and cover change (LUCC) data: the compositions of different vegetation and non-vegetation in each grid-cell, including forest, cropland, grassland, shrubland, wetland, barren land, water body, glacier, and impervious land. **For historical period**, the distribution of water body and glacier areas was derived from the GlobeLand30 produced by the Ministry of Natural Resources of China (<https://lcviewer.vito.be/download>). The impervious land was directly resampled from published data covering the period 1978-2017 (Gong et al 2019), while the periods 1900-1977 and 2018-2019 were assumed to be the same as in 1978 and 2017, respectively. Other vegetation cover types, including cropland, forest, wetland, grassland, and shrubland were reconstructed individually. To do so, we developed a top-down model to reconstruct the historical distribution of cropland, forest, and wetland in China spanning the period 1900-2019. The model allocated a specific, prior-determined, provincial-level area of different biomes to grids in China. The LUCC database assimilates land conversion signals from reports, field surveys, and satellite images (Yu et al 2022). **For future period**, the forest expansion was considered according to the official forestation plan. Specifically, new forests will be established at 15 Mha from 2016 to 2020 and at an additional 49.5 Mha from 2021-2050. Since China restricts further conversion of croplands and wetland to forestland for food security and biodiversity conservation, we assumed that the future plantations were only

allowed to be converted from grassland and shrubland. More details were provided in Lines 596-623.

The above information was mostly provided in Material and Methods section and the details of the historical LUCC were described in Yu et al (2022). Please check Lines 406-455 and 711-718.

3) Confusion between recent trend and future projections:

We apologize for any confusions caused. Our results presented were majorly focused on future projection (i.e. carbon potential). We have tried our best to avoid such confusions by specifying the periods. Please check our revisions in Lines 58, 249, and 255-256 for clarification. We hope these revisions are competent.

4) Uncertainties:

In this study, we spotted a few sources that caused the large uncertainties reported in previous studies. We examined the methods, forcing data, and assumptions adopted in these studies and found that there were inappropriate (i.e. unrealistic) designs in some of the studies. **Therefore, our “reducing uncertainty” means giving projections that more close to realistic conditions.** In other words, we reduced the uncertainties in the previous reported studies by providing more reasonable estimates. We did this by addressing the uncertainty sources through rigorously calibrating and validating of carbon pools, avoiding the unrealistic assumptions, and improving the simulation methods. Here are three examples that help reduce uncertainties:

- a) We found that previous studies were largely diverged in projection of carbon stock changes, while the studies that were calibrated using NFI data were much more converged. Therefore, in our study, we carefully and intensively calibrated and validated both carbon stock and carbon sink during the historical periods (see Figure S14). Moreover, studies with abnormally high initial (e.g. in 2020) forest carbon stock are prone to report lower carbon accumulation potential as the carbon saturation is ecologically and biologically predefined in model simulations. Thus, our calibration and validation help position the carbon stock of each forest types at a correct level (initial status for projection of future carbon stocks).

- b) We avoided unrealistic assumptions been adopted in previous studies. For example, forestation failures are common but often ignored in carbon stock projections. If forestation failures were ignored, the biomass carbon accumulation from forestation will be overestimated. Here, we considered forest failure by assuming different tree survival rates in projections using both statistical and process-based models. More specifically, the baseline tree survival rate was set to 47%, which was derived from the 9th NFI in the arid and semi-arid region of Northwest China. While in comparison, an elevated tree survival rate was set to 85% (the threshold above 85% qualifies for forestation as set by the State Forestry Administration of China) assuming improved forest management.
- c) Improvements in process-based simulations using DLEM. We made many improvements to force our simulation close to realistic condition. For example, for future forestation, the forest type or tree species used were difficult to be determined. Studies using different forest types or tree species for carbon stock projections will result into largely diverged estimations. In this study, we considered the dominant forestation tree species in each province, as well as climate change impacts on land suitability. By limiting the species-specific land suitability in future forestation, our projection of carbon accumulation in new plantations was also constrained and more realistic. The other example that warrants our simulations more realistic is the consideration of forest harvesting. Limited by the deficit of statistical model, forest harvesting was absent in this approach. While for process-based model, representing forest harvesting will require forest demographic information. Leveraging by the forest demographic information derived from the National Forest Inventory (NFI) survey data, we were able to address this challenge for the first time.

The above are the three typical examples that reduce the uncertainties by giving a more realistic projection of carbon stock in comparing to former studies. Besides, our other improvements also help to yield a more realistic estimation, including limiting forest harvest in timber forest only, taking into account forest mortality under

changing climate, and forest age increment adjustments (i.e. annual stand age accrual does not equal to 1). To summarize, our reducing uncertainty means giving projections closer to realistic conditions. However, we could re-examine removing the descriptions about “reduced uncertainty” from the text if the reviewer insists that these statements are not beneficial. If so, please kindly let us know.

References:

- Gong, P., Li, X. & Zhang, W. 40-Year (1978–2017) human settlement changes in China reflected by impervious surfaces from satellite remote sensing. *Sci. Bull.* **64**, (2019).
- Jia, Y. *et al.* A spatial and temporal dataset of atmospheric inorganic nitrogen wet deposition in China (1996–2015). *Sci. Data Bank* **4**, 76–83 (2018).
- Salas-Eljatib, C., Mehtätalo, L., Gregoire, T. G., Soto, D. P., & Vargas-Gaete, R. (2021). Growth equations in forest research: mathematical basis and model similarities. *Current Forestry Reports*, 1-15.
- Yu, G. *et al.* Stabilization of atmospheric nitrogen deposition in China over the past decade. *Nat. Geosci.* **12**, 424–429 (2019).
- Yu, Z., Ciais, P., Piao, S., Houghton, R. A., Lu, C., Tian, H., ... & Zhou, G. (2022). Forest expansion dominates China’s land carbon sink since 1980. *Nature communications*, *13*(1), 5374.

Hereafter I provide some detailed remarks following the text:

Abstract

General remark : Please, better define the distinction between potential C sequestration in forest biomass and what it would take to reach this potential in terms of forest management practices under various RCP scenarios.

Response: We thank the reviewer for the insightful suggestion. As the journal sets words limit of 150 for abstract, and the current version exceeds the limit by 268 words. There are many contents need to be truncated thus impeding to explain this in detail. However, we agree this

should be explained and we hereby revised Lines 45-48 to remind readers that improved forest management practices will be the key to achieve the goal.

Line 36-37 : « Our projections indicate a remarkable increase in biomass carbon stock », under which conditions/assumptions ?

Response: We thank the reviewer for the suggestion. This is an average number derived from simulations assuming two survival rates (i.e. 47%, 85%) in four SSPs (i.e. SSP1-2.6, SSP2-4.5, SSP3-7.0, and SSP5-8.5), and we have added this information in Lines 40-42.

Line 37-38 : same remarks, what would it take to reach this biomass C stocks in 2100 (no harvest ? sustainable harvest ? under which RCP scenario ?)

Response: We thank the reviewer for the suggestion. As explained previously, the current abstract exceeds the words limit, and we added the information at the end of the abstract. The revision is to remind reader that improved forest management practices will be the key to achieve the goal. Please check the revision in Lines 45-48.

Introduction

Lines 53-62 : Here again, it is a bit confusing to mention the uncertainties regarding the estimates of C stocks and sink in forests together with the lack of « optimal forest management plan and strategies ». Please differentiate the research-gap associated to the monitoring of Forest C stocks and sinks (recent trend) and the research-gap regarding the assessment of the potentiel C sequestration in forests. These are two different problems which should be clearly distinguished even though they are linked to one another.

Response: We thank the reviewer for the constructive comments. We agree and rephrased the sentence in Line 60. Moreover, this study majorly focuses on assessment of the carbon sequestration potential (long-term projection). Therefore, we made revisions throughout the text to avoid the confusion. Please check our revisions in Lines 58, 249, and 295.

It is also not clear if the author are focusing on forest biomass C stocks or forest C stocks (including biomass, dead wood, litter, soil). Please specify.

Response: We thank the reviewer for capturing this. In this study, we focused on forest biomass C stock, and the dead wood, litter, and soil carbon were excluded. To make this clear, we rephrased the sentence in Line 74 for clarity.

Line65-70 : So the time-frame of the study is 1999-2018 ? I do not understand what are the objectives of the study. Is it to estimate the forest C sink in forest (biomass ?) over the 1999-2018 period in China ? Please present more clearly : (i) the resarch-gap(s), (ii) the research question(s), (iii) the research objective(s). In its current form the introduction is too elusive and does not enable to understand the purpose of the research.

A brief presentation of the method should be presented at the end of the introduction to understand what has been performed. This would be the end of the introduction (iv) how these objectives were reached.

Response: We thank the reviewer for the constructive comments. This study focused on projection of forest biomass carbon potential in the period of 2020-2100, while the National Forest Inventory (NFI) survey data during 1999-2018 were used to calibrate and validate the model. To make this clear, we rephrased the sentence in Lines 74-75.

Moreover, we have rephrased the Lines 69-73 to clarify the research gaps, research questions, and research objectives. Here are the details:

“However, the current research gap mainly lies in the lack of collective considerations of the above key factors, which further constrain previous assessments of China’s forests’ carbon-holding capacity. Therefore, this study aims to: 1) address these challenges in an integrated manner for more accurate projection of forest biomass carbon potential; and 2) assess the improved management practices to enhance carbon potential.”

Besides, we also added a brief description of the methods used at the end of the introduction as suggested to explain how the objectives were reached:

“We employed a combination of both statistical and process-based biogeochemical models for carbon stock and sink projection. Specifically, for simulations using process-based

model, we designed four groups of experiments, in which the first group is to quantify the carbon dynamics without improved management practices, and the second group is to quantify the additional carbon stock and sink from improved forest managements. While in comparison, the third group experiments were used to quantify the bias introduced by neglecting of wood harvest, and the fourth group was to quantify the sink contributions from existing and new forests. In this study, the optimal managements are the carbon stock and sink enhancement practices, including the replacement of inappropriate tree species by indigenous species, and reduction of the harvesting intensity by postponing the harvesting time (see methods).”

Results and Discussion

Lines 73-74 : Unclear what has been performed. If wood harvest has been derived from forest growth and regrowth, this should be presented as a result and compare with independent data on forest wood harvest, either from national statistics or at least from the FAO database (at the national level).

Response: We thank the reviewer for the comment. In our simulation, we optimized and determined the timber harvest by species and age, and the harvesting impacts on forest age further affect the forest demographic and growth. In former studies, timber harvest was generally represented as carbon removed from gridcells in the unit of fraction (% of total carbon) or unit of kg C per year in simulations using process-based model. A typical example is to use the LUH2 harvesting data in model simulations. The LUH2 provides future wood harvest information (https://luh.umd.edu/LUH2/LUH2_v2f_README_v6.pdf), which were standardized into a total national wood harvest demand (Mg C yr^{-1}) (Hurtt et al. 2020). Therefore, the wood harvest was demand-driven and using the latest year as baseline for spatial disaggregation. However, since wood harvest is species-specific, in which the harvesting cycles (i.e. rotation length) will be differed by locations and further perplexed by forestation, tree species replacement. Moreover, in China, a substantial portion of forests were banned from harvesting, because they were established in ecological fragile region for environment protection, such as sand fixation, soil erosion prevention, and biodiversity conservation. These were also

considered in our simulations by limiting wood removal in timber forests only, which is more realistic.

Therefore, in our simulation, we improved the forest harvest processes by considering tree species and forest age. Moreover, the harvesting impacts on forest age were further considered in simulations, which will affect the forest demographic and growth. These mechanisms improve the carbon removal and accumulation processes in simulations.

The carbon harvested from forest was 55 Tg yr⁻¹ during 2000-2020 based on our simulations, which is in the range of 40 -100 Tg C yr⁻¹ obtained from other studies during 2003-2018 (FAO, national statistics, LUH2, and Wang et al 2022). We added this information in Lines 179-181. Besides, we have also added some of the above information in the text to increase the readability. Please check our revisions in Lines 94-100.

References:

- FAO: Forestry data, FAOSTAT Database, Food and Agriculture Organization of the United Nations, Rome, Italy, Available at: <https://www.fao.org/faostat/en/#data/FO>, 2020, last access: 25 June 2023.
- J., Kaplan, J. O., Kennedy, J., Krisztin, T., Lawrence, D., Lawrence, P., Ma, L., Mertz, O., Pongratz, J., Popp, A., Poulter, B., Riahi, K., Shevliakova, E., Stehfest, E., Thornton, P., Tubiello, F. N., Van Vuuren, D. P., and Zhang, X.: Harmonization of global land use change and management for the period 850–2100 (LUH2) for CMIP6, *Geosci. Model Dev.*, 13, 5425–5464, <https://doi.org/10.5194/gmd-13-5425-2020>, 2020.
- Wang, D., Ren, P., Xia, X., Fan, L., Qin, Z., Chen, X., and Yuan, W.: National forest carbon harvesting and allocation dataset for the period 2003 to 2018, *Earth Syst. Sci. Data Discuss.* [preprint], <https://doi.org/10.5194/essd-2023-309>, in review, 2023.

Lines 75-76 : Suddently we move from recent forest C sink to future forestation. Again, the data frame is very obscure. What is the temporal projection ? 2100 ?

Response: We thank the reviewer for the comment. We are sorry for the confusion. The descriptions about harvesting and forestation species selection were improvements served for

forest biomass carbon projection. To avoid the confusion, we have added more information and rephrased the content. Please check our revisions in Lines 94-103, 117 for more details.

Lines 77-78 : What are the studies using « initial carbon stock validated by inventory data » that you are mentioning here ?

Response: We thank the reviewer for the comment. These are the studies of Qiu et al (2022), Zhang et al. (2018), Li et al. (2018), and Hu et al. (2015). We found that the modeled carbon stocks were validated using national forest inventory data in these studies. We have added this information (references) in Line 106.

References:

Hu, H., Wang, S., Guo, Z., Xu, B., & Fang, J. (2015). The stage-classified matrix models project a significant increase in biomass carbon stocks in China's forests between 2005 and 2050. *Scientific Reports*, 5(1), 11203.

Li, Q., Zhu, J., Feng, Y., Xiao, W. (2018). Carbon storage and carbon sequestration potential of the forest in China. *Climate Change Research*, 14(3), 398-294. doi:10.12006/j.issn.1673-1719.2017.102

Qiu, Z., Feng, Z., Song, Y., Li, M. & Zhang, P. Carbon sequestration potential of forest vegetation in China from 2003 to 2050: Predicting forest vegetation growth based on climate and the environment. *J. Clean. Prod.* **252**, (2020).

Zhang, C., Ju, W., Chen, J., Fang, M., Wu, M., Chang, X., ... & Wang, X. (2018). Sustained biomass carbon sequestration by China's forests from 2010 to 2050. *Forests*, 9(11), 689.

Line 84-86 : Which process-based model did you use ?

Response: The model used was DLEM, and we added this information in Line 114.

Lines 86-89 : What are the tree survival rate scenarios ? How were they established ? What kind of statistical model did you use ? Which trajectories of wood harvest, CO2 conc and climate

change did you implement ? How did you choose them ? Even though the methods are at the end of the paper, what you have performed should be clearly stated before each results.

Response: We thank the reviewer for the comment. Here are the details to each question:

- 1) survival rate scenarios: Most of the previous studies assumed that all new forestations were successful, which is not realistic. Here, in both simulations using statistical and process-based models, we consider forestation failures by constraining tree survival rates between harsh conditions (e.g. arid and semi-arid regions) and the expected criteria designed by the State Forestry Administration. Specifically, for new planted forests, simulations were performed assuming a baseline tree survival rate (47%, derived from the 9th NFI in the arid and semi-arid region of Northwest China) and a scenario with elevated tree survival rate (85%, the threshold above which land qualifies for forestation as set by the State Forestry Administration of China) under improved forest management.
- 2) statistical model: the statistical models used were carbon stock growth models. We established stand-level carbon stock growth models for forests with differed origins (i.e. planted and natural forests), site classes (i.e. 3-9 site classes based on the regional distribution of species and the number of sample plots), and provinces (i.e. 31 provinces in total), using three theoretical growth equations (i.e. Richard, Korf, and Hossfeld). These models were carefully validated using measured data, and the one demonstrating the highest performance was selected for prediction of future carbon stock, incorporating crucial factors such as forest age, stand class, and tree densities.
- 3) wood harvest, CO₂ concentration and climate change: In process-based (DLEM) simulations, we designed multiple experiments which can be divided into four groups: the first group is to quantify the carbon stock and sink changes without optimized management practices, and the second group is to quantify the additional carbon stock and sink if forest management is optimized. The third group of experiments were designed to examine the bias if wood harvest is neglected, and the fourth group as to quantify the sink from existing and new forests. Therefore, wood harvests were implemented in group 1, 2&4 simulations, and group 3 was designed to be free from

harvesting. These information has been added in the last paragraph of the introduction (Please see Lines 80-90). Besides, CO₂ and climate change were varying and represented in the forcing data of each SSPs. We also added this information in Lines 129-131.

To sum up, we agree and have rephrased the paragraph to provide the information in the results and discussion section as suggested. Note that some of the information has also been added in the introduction in response to previous questions. Please check our revisions in Lines 80-90 and 117-131.

Line 89 : What are those « vast majority of other studies » that you are referring to ?

Response: We thank the reviewer for pointing this out. As far as we know, none of the studies listed in the Figure 1 have considered forestation failure specifically. These studies assumed that all new forestations were successful, which is not realistic. To make this clear, we added some of the references here. Please check our revisions in Line 119.

Lines 104-106 : Will increase to 21.6-24.3 PC in 2100 under which assumptions of climate change, afforestation, wood management ? It is also very obscure how did you reduce the range in uncertainties compared to other studies. The approach implemented to reduce the uncertainty range should be stated very clearly. It is really unclear how you validated your models, knowing that model simulations can ONLY be validated against independent observation time-series data.

Response: We thank the reviewer for the suggestion. The range was derived from statistical and process-based models. Specifically, for statistical model simulation, harvesting was not considered (not able to be considered due to model limitation). While for the process-based model simulations using DLEM, the results were derived from the first group simulations, in which a total of 8 simulations were conducted assuming two tree survival rates and four SSPs (Please see details in Table S5). We added information in Lines 138-140 for clarification.

For future projections, there is no observation data available for validation. Therefore, the validations were performed for historical period only. We spotted a few sources of

uncertainties by examining the methods, forcing data, and assumptions in previous studies. We reduced the uncertainties by addressing these uncertainty sources through rigorously calibrating and validating of carbon pools, avoiding the unrealistic assumptions, and improving the simulation methods. We have provided three examples that help reduce uncertainties in the responses to the first questions. **Please kindly let us know if the reviewer thinks that “reduced uncertainty” is still not beneficial, to re-consider removing the descriptions from the text.**

Lines 113-129 : This whole passage remains completely obscure as long as we don't know which models were used, which are the controlling (inputs) variables of the models, following which assumptions were the value of these input data set, and, most importantly, how the uncertainties of model input AND output data were estimated. Again these uncertainties can, by no way, be taken as the envelope of the model results as long as the models have not been first evaluated against observational data.

Response: We thank the reviewer for pointing this out. There are only two types of models used in our estimation, i.e. statistical model and process-based model. For statistical model, three theoretical growth equations (i.e. Richard, Korf, and Hossfeld) were applied for each forest type, and the one demonstrating the highest performance was selected for carbon prediction. For process-based model, we used DLEM to simulate the carbon. The results of Figure 1-3 were derived from the eight experiments in simulation group 1. The details of the four simulation groups have now been explained in the last paragraph of Introduction. We also iterate this in Lines 151-152 and 155-162 to remind readers.

Our simulations using DLEM model have been intensively calibrated and validated for both carbon stock and sink in the historical period (please see the species-level validations in Figure S14). The model assumptions and input have now been explained in previous text in Lines 117-131, 138-140, 151-162. We also added explanations in Figure legends to make this clear for reader.

Line 127-128 : No timber harvest does not promote forest C sequestration, it only rejuvenate forests so that there is obviously a C sink due to forest regrowth. But this forest regrowth can be much less efficient than letting forests age as much as possible. Because even though the C sink may diminish with forest ageing, the important is not the C sink but the C stocks which transit between forest ecosystems – timber goods – the atmosphere !

Response: We thank the reviewer for pointing this out. We agree with the insightful opinion raised by the reviewer. If forest was free from harvesting, the carbon will be held in the ecosystem, despite the sink will diminish with aging. From a long-term perspective, if all forests were free from harvesting, the climate mitigation effect will be larger when comparing with scenario of harvesting. However, totally banning of forest harvesting is impossible, while our simulations were targeting at projecting carbon dynamics as close to realistic as possible.

Lines 140-141 : which categories of wood products have been distinguished ?

Response: The wood products were categorized into short-, mid-, and long-live types (i.e. paper/paperboard, wood-based panels, and sawn wood, respectively). We have added this information in Lines 182-183.

Figure 1. I do not understand how there can be « inventory-validated studies » for 2030-2060 (cyan box panel a) ? So the range of uncertainties reflect the divergence between various scenarios of tree survival rate and other « different scenrios » for the process-based model ? This is really not an uncertainty estimates but a sensitivity analysis to different choice. It is an overstatement to say that the present research enable to reduce uncertainties regarding forest C sink in China.

Response: We thank the reviewer for the comment. The inventory-validated studies were studies that used national forest inventory data to validate their models before projection. We have added this information in the figure legend. Please check our revision in Lines 193-194.

We also want to clarify that we did not simply use the combinations of different factors in our simulations, i.e. they were not simply a sensitivity analysis. Alternatively, our experiments were purposely designed to consider relatively extreme but close-to-realistic

conditions. For example, the overall tree survival rate in new plantations will most likely fall between the 47-85%, and the future greenhouse gas concentration and climate change will most possibly follow the changes between SSP1-2.6 and SSP5-8.5. Instead, we claimed that our study reduced the uncertainties because our reported range was more realistic and was much converged in comparing with the large variations reported in other studies (i.e. the grey box). We kindly recommend the reviewer to reconsider our statements based on the above and previous explanations. **However, as we previously explained, we are willing to re-examine removing the term “reduced uncertainty” if the reviewer insists that the statements are not beneficial. Please kindly let us know.**

Lines 159-167 : Please provide a clear definition of non-timber natural forest », « timber natural forest », « non-timber planted forest », and « timber planted forest ».

Response: We thank the reviewer for capturing this. We have added information in Lines 204-211 to explain this.

Figure 2. It is a bit surprising to see that there is strong spatial overlap of the different forest categories... Could you explain this in the legend.

Response: We thank the reviewer for pointing this out. The figure shows the simulated biomass carbon change in different forest categories. The areas of timber forest for each species were obtained from the China Forest Resource Report 2014-2018. We suspect that the strong spatial overlap of different forests was due to spatial resampling. The maps were resampled from 1 km× 1 km to half degree (~50 km × 50 km) resolution, and therefore different forest categories were mixed and overlapped. We have added the information of timber forest in the figure legend. Please check Lines 225-226.

Lines 174-176 : Please stay with Pg and not change unit.

Response: We thank the reviewer for the comment. Revision has been made as suggested in Line 233.

Line 184 : Forest areas in previous reports were either much smaller than what they are in reality. But we do not know what the reality of forest area will be in 2060. This simply means that you had a more optimistic assumption regarding forest area expansion in this study.

Response: We thank the reviewer for pointing this out. We understand the reviewer's concern. As we explained before, we were trying to project future carbon stock based on conditions that are as close to reality as possible. Therefore, we forced the changes of forest area strictly following the official plan. We believe this constraint is more reasonable and realistic because of two reasons.

First, estimations in former studies are divergent from reality because they either unconsidered future forestation or overestimated forestation. For example, Ju et al (2007) and Zhou et al (2013) simulated forest carbon change in existing forest only. For studies that considered future forestation activities, future forest areas were overestimated (not underestimation), because they aimed at giving a theoretical carbon potential (e.g. 225 Mha increment from 2018 to 2060 by Zhang et al 2022; 78 Mha forestation area by 2060 in Xu et al 2023). Despite that the assumed maximum forest area (e.g. only consider climatic limitations) could be larger than the officially released forest area, it will be unrealistic because the economic limitations were not considered, leaving the maximum forest area a theoretical but unachievable goal. In China, forestation activities are always determined by the official department (i.e. State Forestry and Grassland Administration of China, former State Forestry Administration of China). Therefore, the investments to annual forestation activities in China, determined by the Forestry Department, are planned and implemented by trading the cost and labor. Thus, it is more realistic to constrain the forest area changes by the official forestation plan.

Second, although we do not know the exact forest area in 2060, we are confident that the forest area will be close to the officially released value because the historical forest area changes strictly followed the official plans. For example, the State Forestry Administration of China released "The Twelfth Five-Year Plan" in 2011 to increase forest coverage to 21.66% in 2015. The government almost achieved this goal to increase the forest coverage to 21.63% during the 9th NFI (i.e. 2014-2018). For a more recent period, the official forestation plan (i.e.

the National Forest Management Plan 2016-2050) was to expand forest area by 15 Mha from 2016 to 2020. In comparison, the officially released forest area increased from 207 Mha in the 9th NFI to 220 Mha in 2020 (see: <http://env.people.com.cn/n1/2021/0605/c1010-32123047.html>), which is close to the official plan.

Thus, we believe our forestation plan derived from the officially released document is not an optimistic assumption, but a more realistic assumption compared to former studies.

References:

Ju, W. M., Chen, J. M., Harvey, D. & Wang, S. (2007). Future carbon balance of China's forests under climate change and increasing CO₂. *J. Environ. Manage.* **85**, 538–562.

Xu, H., Yue, C., Zhang, Y., Liu, D., & Piao, S. (2023). Forestation at the right time with the right species can generate persistent carbon benefits in China. *Proceedings of the National Academy of Sciences*, *120*(41), e2304988120.

Zhang Y., Li X., Wen Y. (2022). Forest carbon sequestration potential in China under the background of carbon emission peak and carbon neutralization[J]. *Journal of Beijing Forestry University*, *44*(1): 38–47.

Zhou, L. *et al.* (2013) Carbon dynamics in woody biomass of forest ecosystem in China with forest management practices under future climate change and rising CO₂ concentration. *Chinese Geogr. Sci.* **23**.

Line 186 : Historical forest maps were rigorously calibrated and validated. This should be presented before since model validation is also a result. Therefore, it should come in the result section and a brief explanation of how this was performed need to be presented even though the details are further developed in the methods.

Response: We thank the reviewer for the suggestion. The calibration and validation of the historical forest maps were conducted in our previous study (Yu et al 2022). Therefore, we did not elaborate it here. However, we agree that the explanations of the calibration and validation will be helpful for reader. Therefore, we added the information in the main text. Please check our revisions in Lines 204-211.

References:

Yu, Z., Ciais, P., Piao, S., Houghton, R. A., Lu, C., Tian, H., ... & Zhou, G. (2022). Forest expansion dominates China's land carbon sink since 1980. *Nature communications*, 13(1), 5374.

Line 197 : If timber forest are permanently rejuvenating, the C sink is also permanently compensated by a C source, so that the C sink is not persistent.

Response: We thank the reviewer for capturing this. Indeed, if the timber harvested from forests was immediately burned, the carbon will be directly released into atmosphere.

However, the carbon in wood product pool will decay with different rates depending on product categories. If the ecosystem is treated as a system, there would be persistent carbon sink. However, if the system boundary extended to cover ecosystem and wood product pool, then the carbon balance will be determined by both ecosystem uptake and the wood product decay. We agree that this statement might cause misunderstanding, and we revised the sentence to “Specifically, timber forests routinely harvested will remain relatively young to uptake carbon, ...”. Please check our revision in Lines 255-256.

Line 210-212 : This statement is largely incorrect, except if timber harvest is used for very long-term wood products. Please see Erb, H., Haberl, H., Le Noë, J., Tappeiner, U., Tasser, E., Gingrich, S., (2022). Changes in perspective needed to forge “no-regret” forest-based climate change mitigation strategies. *Glob. Change Biol. Bioenergy*. 12921. DOI: 10.1111/gcbb.12921

Response: We thank the reviewer for pointing this out. We rephrased the sentence to avoid misunderstanding. Please check our revision in Lines 269-271.

Figure 3. First time that the SSP are mentioned... Were they implemented in the previous figures shown?

Response: Yes, the results in Figure 1-3 were derived from simulations with different SSPs. In the revised version, we have mentioned SSP and explained the simulation settings in Lines 129-131.

Line 268: 43.9-166.8 TgC accumulated from 202-2100, this large range of values is in contradiction with the results shown in Fig. 1a?

Response: The carbon accumulation by 43.9-166.8 Tg during the period of 2020-2100 is equal to 0.044-0.167 Pg, which is relatively small when comparing to the stock range of 2.7 Pg (21.6-24.3 Pg) in Figure 1a.

Table 1. Please use the same unit everywhere, this is very confusing otherwise. Are you referring to SSP or RCP? As far as I understand it, SSP are socio-economic pathway while RCP are scenarios of C emissions. Of course, both can be coupled but it seems more adequate to refer to RCP rather than SSP in this case.

Response: We thank the reviewer for suggestion. We have revised Table 1 to keep Tg as the unit used. For the forest management improvement in Table 1, we used Tg because the contributions of some of the practices to carbon change are relatively small (e.g. 0.00017 Pg).

SSP-based scenarios combine elements from the new narratives about future societal development (the SSPs) with the previous iteration of scenarios, the Representative Concentration Pathways (RCPs), which describe trajectories of change in atmospheric GHG and aerosol concentrations (and corresponding changes in radiative forcing) over time. More specifically, SSP based scenarios further refine the previous greenhouse gas concentration scenarios in RCPs. RCPs were explicitly designed for the climate modelling community to explore the effects of different emissions trajectories or emissions concentrations (resulting in various Radiative Forcing values). The socio-economic characteristics used to define RCPs are not standardized, making it difficult to map societal changes like population, education, and government policies to climate targets, such as keeping global warming well below 2°C. SSPs address this by defining how societal choices can lead to changes in Radiative Forcing by the end of the century. As such, SSPs expand on RCPs to allow for a standardized comparison of society's choices and their resulting levels of climate change. Also, the latest iteration of scenarios, used for CMIP6 (2016-2021) and featured in the IPCC Sixth Assessment Report (AR6) (2021), are based on a set of Shared Socio-economic Pathways (SSPs). Therefore, we chose to use SSPs in our simulations.

Method

Lines 324-325: What are those equations. On which variables do they rely and how did you get access to these variables? This is really too elusive and we can't understand what has been performed here

Response: We thank the reviewer for the comment. We have double-checked and confirmed that the three models were all used for all tree species/species groups in carbon stock predictions. Then, the average of the predictions from the three models were used using R^2 as the weighting factors. We have added this information in the Method section. Please check our revisions in Lines 390-404. The details of the equations used have been summarized in the supplement excel file. Please also check the file for more details.

Line 328-331: Which data were used to validate the models? Saying that careful validation was performed is really not enough and not convincing at all.

Response: We thank the reviewer for the comment. The data used to validate the model was the National Forest Inventory data. For each tree species/species group, the forest plots were divided into ten parts, model parameterizations were conducted by nine times with each of ten parts been used as validation samples. The one with the highest performance was selected for prediction from each of the Richard, Korf, and Hossfeld model-group. Then, the predictions from the three models were averaged using R^2 as the weighting factors. We have added this information in the main text. Please check our revisions in Lines 390-404.

Lines 333-337: But which process-based model did you use? What are the equations of the model? What input variable are required to run it?

Response: We thank the reviewer for the comment. The process-based model used was DLEM, and we have added this information in Line 459. The DLEM is a typical terrestrial biospheric model (see Figure 2 below), which estimates carbon fluxes through a set of physical and physiological relationships based on our current understanding of processes controlling carbon exchange. More specifically, the DLEM is a highly-integrated, process-based ecosystem model

(see Figure 3 below) that aims at simulating the structural and functional dynamics of land ecosystems affected by multiple factors including climate, atmospheric compositions (CO₂, nitrogen deposition), land use and land cover change, and land management practices (harvest, rotation, fertilization etc). The DLEM has five core components: 1) biophysics, 2) plant physiology, 3) soil biogeochemistry, 4) dynamic vegetation, and 5) land use and management. The model operates at a daily time step with spatially explicit information on climate, vegetation and soils to provide estimates of carbon, nitrogen and water fluxes and pools of terrestrial ecosystem from site to regional to global scales. Therefore, the model cannot be simply described by a set of equations.

The input variables needed to force the model have been elaborated in the “Climatic and atmospheric chemical condition datasets”, “Experimental designed for model simulations”, and “LUCC data and model validation” subsections, as well as the descriptions in the Lines 129-131. However, we realized that it will be more logical to introduce “Climatic and atmospheric chemical condition datasets” and “Experimental designed for model simulations” before “Age impacts on forest carbon uptake in process-based model”. Therefore, we restructured this part to make it more clear to reader. Please check our revisions in Lines 406-455.

Figure 2. The model structure of DLEM. Sources: “Figure 1 The simplified framework of Dynamic Land Ecosystem Model (DLEM) for assessing the effects of climate change and increasing

atmospheric CO₂ concentration on global terrestrial net primary production (NPP).” by Pan et al (2014) is licensed under CC BY 4.0 (<https://creativecommons.org/licenses/by/4.0/>), no change was made.

Figure 3. The carbon and nitrogen processes in DLEM. Sources: “Fig. 1 Nitrogen processes and C–N coupling in the dynamic land ecosystem model (DLEM)” by Lu et al (2012) is licensed under CC BY 4.0 (<https://creativecommons.org/licenses/by/4.0/>), no change was made.

Reference:

Chaoqun Lu, Hanqin Tian, Mingliang Liu, Wei Ren, Xiaofeng Xu, Guangsheng Chen, Chi Zhang.

Effect of nitrogen deposition on China's terrestrial carbon uptake in the context of multifactor environmental changes, *Ecological Applications*, 22(1): 53-75, 2012.

Pan, S., Tian, H., Dangal, S. R., Zhang, C., Yang, J., Tao, B., ... & Li, X. (2014). Complex spatiotemporal responses of global terrestrial primary production to climate change and increasing atmospheric CO₂ in the 21st century. *PLoS one*, 9(11), e112810.

Lines 351-390: It is difficult to understand which data were used to implement the age

correction factor and how this could be species-specific. If I understand well authors have optimized the a and b parameters based on the C sink simulated by the process-based and statistical mode. But this sounds very tautological because the a and b parameters should be implemented within the process-based model. How is this validated against observation data?

Response: We thank the reviewer for the comment. We are sorry for the confusion. In this study, we divided forest into four groups, including natural timber forests, natural non-timber forests, planted timber forests, and planted non-timber forests. Each of the group consists of 23 species/species group at most as listed in Table S6. The parameters a and b were developed for each species/species group, and were applied in process-based model (i.e. DLEM) simulations only (not for statistical model).

As previously explained in the responses to the first question, the parameters were developed based on forest quantitative theory. As showed in Figure 1a in this response file, trees grow relatively slowly at first, increasing their growth-rate to the point of inflection of the growth curve. In phase 2, the growth is fast, and in phase 3, there is a decrease of the growth-rates.

According to the forest quantitative theory (and adopted by the State Forestry Administration of China), the wood harvest will be performed at t^b in Figure 1b in this response file (i.e. the year of highest accumulated growth rate) to obtain the largest profit. Therefore, we used the harvest age data to calculate the parameters a and b. The harvest ages for each species/species group were determined by the State Forestry Administration of China based on decades of experiences and observational data obtained from field. Therefore, the parameters a and b were derived from an official document that has solid basis.

Please also check more details in our response to the first question.

Lines 393-399: Which data have been used to implement the biophysics, physiological and biogeochemistry processes: what are the input data and main parameters?

Response: We thank the reviewer for the comment. The DLEM was driven by climate, atmospheric compositions (CO₂, nitrogen (N) deposition), land use and land cover change, and land management practices (harvest, fertilization etc). As we explained previously in Figure 2 in

this response file, the biophysics, physiological and biogeochemistry processes are integrated and related. We have restructured the Method section to explain the forcing data used in DLEM before the improvement of the model. We believe these changes will be helpful to understand the input data and key parameters in the simulations.

Please also check our previous response (i.e. response to “Lines 333-337: But which process-based model did you use? What are the equations of the model? What input variable are required to run it?”) for more details.

Lines 420-430: Tree mortality may also depend on forest age but ok.

Response: We thank the reviewer for insightful comment. Indeed, the tree mortality also related to forest age. We did examine the relationships between tree mortality and age, but a general relationship was not identified. To reduce the complexity (as we have considered many factors in simulations), we simply randomized the tree mortality and related it to climate (i.e. SSPs). This could be further considered in future studies for improvement.

Lines 481: 15Mha from 2016 to 2020: has this really occurred?

Response: We thank the reviewer for capturing this. According to the officially released data, the forest area has increased from 207 Mha in the 9th NFI to 220 Mha in 2020 (see: <http://env.people.com.cn/n1/2021/0605/c1010-32123047.html>), which is close to the official plan of 15 Mha.

Lines 505-508: to me this sounds more like RCP rather than SSP scenarios. Maybe correct

Response: We thank the reviewer for the comment. The climate data were obtained from the Copernicus Climate Data Store (<https://cds.climate.copernicus.eu/cdsapp#!/dataset/projections-cmip6?tab=overview>), where they were described as SSP data.

Lines 561-562: But we do not understand how the calibration and validation have been performed

Response: We thank the reviewer for pointing out this. The historical calibration and validation were conducted in our previous study (Yu et al 2022), and we also conducted a species-level validation using the carbon stock and sink derived for the most recent NFI period (i.e. the 9th NFI period from 2014 to 2018). Specifically, the DLEM simulated carbon stock was validated using NFI data. Besides, we used the carbon sink derived from statistical model (i.e. growth model) as a reference to validate carbon sink derived from DLEM simulations during the period of 2014-2018. This is because we believe that the carbon sink during a relatively short-term period can be accurately captured by growth models. Please check our revisions in Lines 429-431.

Supplementary materials

What is the source of the parameters presented in Tables S2-3?

Response: The sources of the parameters in Table S2 can be found from the references listed in the table. The parameters in Table S3 were obtained from different references and we have added this information in the revised version. Please check our revisions in Table S3.

REVIEWER COMMENTS

Reviewer #1 (Remarks to the Author):

I reviewed the manuscript the second time. The revised version have made some clarifications mainly for results and methods. However, the experiment scenarios and parameter settings are still confusing, and therefore so are the results. It is not clear which are the findings with and without (what) optimal forest management. As this is the main conclusion that the paper aims to make, these must be very clear. The definition of optimal or improved forest management is still not clear either. Is wood harvest not included in forest management? Also, what is forest demographic changes, they all need clarifications.

This work is nothing to do with "reducing uncertainty". By comparing to other studies, it could say that the accuracy of estimate is improved, but not reducing uncertainty. It could be proper to change "reducing uncertainty" to "improving accuracy of estimate"/ "closer to the reality" as what was agured in presenting the results.

line 37, not only wood harvest but also species replacement, structural composition of forests. so here "wood harvest" should be "forest mamagement"?

line 38, How did the process-based model correct the bias?

The description of the four experiment groups are not clear in the method.

line 83, "without improved management"? in your simulations, havest is included (Table S5). Is it not a type of management?

line 85, after "improved forest managements", add "i.e., expansion of new forest, wood havest, age rotation, and tree replacement"

line 87, the 4th group, without considering any management?

line 87-90, this definition should be ahead of division of the four groups.

line 94, what is LUH2?

line 117, why wood harvest is emphasized in the paper and other management measures (expansion of new forest age rotation, and tree replacement") are relatively less emphasized? Your simulations have included all of these measures, right?

line 138, "Specifically, the results of DLEM (i.e. process-based model) simulations were derived from group 1 experiments".

What does this sentence mean?

The description of the results is not corresponding to the group settings. I am still confused when reading the results about which have and have not included (what type of) forest management.

line 138-142, is it to compare the results between with and without wood harvest? How did this result come from? "...primary of the sink (85%) is attributed to natural growth and regrowth of forests...".

line 145, "all scenarios with different tree survival rates and SSPs", from which scenarios?

line 151, "Note that DLEM simulations were derived from experiments of simulation group 1 in Figure 1. "What does this sentence mean? Other groups are also simulated by using DLEM.

The previous question is not answered properly. <> , your response is <<64 experiments were conducted>>. If so, this is not a "random" design. What I meant is, whether mortality (47% or 85%) is randomly designed for different cells across space.

line 431, about validation, you stated in the response "the National Forest Inventory (NFI) survey data during 1999-2018 were used to calibrate and validate the model"? but in FigS14, validation was only using data from 2020. What about the other years?

line 444, "wood harvest" is not included in "improved forest management"?

line 450-451, group 1 and group 3 have different settings of CO₂. Therefore, the effect of CO₂ on carbon sequestration is not separated from that of wood harvest. Please explain.

What is the idea of improve tree survival rate from 47% to 85%? How can it be achieved? How do you consider the relationship between tree survival rate and CO₂ concentration?

FigS8, Is harvest included? In FigS10, the results represent scenarios of no harvest and age accrual. What about FigS8?

FigS11, what is the standard/principle for wood harvest setting between 2020 and 2100?

FigS12,13, not clear which scenario did the results represent?

FigS15 How was the location of the new forest determined ? what is the unit of the numbers in the figure?

FigS16 what is the unit of the numbers in the figure, possibility?

Table S5 56 experiments in group 2? What is the total number of all experiments?

Reviewer #2 (Remarks to the Author):

Thank you for taking the time to address my concerns. This is a great study and I continue to be supportive of publication in Nature Communications.

Reviewer #3 (Remarks to the Author):

I thanks the authors for their thorough revision of their manuscript, which, in my opinion, has improved a lot. The authors have answered to all my queries in a satisfactory way.

One thing though, regarding the "reduction of the range of uncertainties" (see my previous comment lines 104-106): I still think it would be more stringent to rather speak of this as an "exploration of realistic sensitivity analysis based on Chinese forestry plan with data-constrained by the 9th NFI integrated into validated models".

But despite this last comment, I can only congratulate the authors for their work on revising the manuscript.

Response to reviewer comments

We thank reviewers for their valuable suggestions to improve the manuscript. We have addressed all the comments raised by the three reviewers. Please find our point-by-point response below.

REVIEWER COMMENTS

Reviewer #1 (Remarks to the Author):

I reviewed the manuscript the second time. The revised version have made some clarifications mainly for results and methods. However, the experiment scenarios and parameter settings are still confusing, and therefore so are the results. It is not clear which are the findings with and without (what) optimal forest management. As this is the main conclusion that the paper aims to make, these must be very clear. The definition of optimal or improved forest management is still not clear either. Is wood harvest not included in forest management? Also, what is forest demographic changes, they all need clarifications.

Response: We thank the reviewer for the suggestions. We have revised the manuscript as suggested. We believe that the confusion mainly caused by: 1) the definitions of the three terms: “forest management”, “improved forest management”, and “optimal forest management”; and 2) the design of the four group of experiments.

Regarding the definitions, we defined the three terms clearly in the introduction. Specifically, **forest management** includes wood harvest and forest expansion, which were represented for accurate projection of carbon dynamic trajectory. The **improved management practices** include the replacement of inappropriate tree species by indigenous species and reduction of the harvesting intensity by postponing the harvesting time. While in comparison, the **optimal managements** indicate the improved forest management practices portfolio that achieves the largest carbon sink. Please check our revisions in Lines 78-85 and 350-351 for details.

Regarding the experiment design, we revised the text to clarify the experiment groups used. Specifically, we added “Note that the results reported were all derived from group 1 experiments, unless otherwise indicated.” at the end of the introduction section. The group 1 experiments were designed to be driven by forcing close to realistic conditions, including forest expansion and wood harvest follows the plans and regulations of the National Forestry and Grassland Administration with environmental factors (i.e. climate and CO₂) varying under different SSPs (see these revisions in Lines 444-449 of the method section). The second group of experiments was designed to account for the impacts of improved forest management practices. While in comparison, the group 3&4 experiments were served as supplement to support the validity or factorial attributions of the results in group 1. Note that group 3 experiments were only used in Lines 164-166 as supplement for comparison, where it is specified, and they were not used in the sink/stock reported.

Besides, “forest demographic changes” indicate the tree age structure changes in forests, which was inevitable due to disturbances such as planting trees, wood harvesting, tree mortality. We have revised the sentence in Line 65 to make this clear to reader.

This work is nothing to do with "reducing uncertainty". By comparing to other studies, it could say that the accuracy of estimate is improved, but not reducing uncertainty. It could be proper to change "reducing uncertainty" to "improving accuracy of estimate"/ "closer to the reality" as what was agured in presenting the results.

Response: We thank the reviewer for the suggestion. We have revised the descriptions of “reducing uncertainty” by replacing the term with “enhance the accuracy of estimates” as suggested in the text. Please check our revisions in Lines 105-106.

line 37, not only wood harvest but also species replacement, structural composition of forests. so here "wood harvest" should be "forest management"?

Response: We thank the reviewer for the comment. Here in the abstract, we highlighted “wood harvest” and “forest demographic changes” because they were neglected in the most of the former studies. However, “forest management” is too broad and it was considered in our

former study published (i.e. Yu et al 2021). Therefore, to keep our statement accurate (i.e. not exaggerated) and to avoid disputation, we think it would be more appropriate to keep “wood harvest” here.

Reference:

Zhen Yu; Weibin You; Evgenios Agathokleous; Guoyi Zhou; Shirong Liu ; Forest management required for consistent carbon sink in China's forest plantations, *Forest Ecosystems*, 2021, 8(1): 8-54.

line 38, How did the process-based model correct the bias?

Response: We thank the reviewer for pointing this out. The process-based model corrected the bias by considering the wood harvest impacts on carbon removal and forest demographic changes. More specifically, wood harvest was applied to timber forests and the tree removal also triggered planting/regenerating of new forests. Such changes in forest demographic further altered forest growth and carbon accumulation processes. However, these mechanisms were unable to be represented and captured in statistical model. To evidence the advantages of the DLEM simulations, we also designed group 3 experiments. The results showed that neglecting wood harvest in DLEM would produce estimations close to the results derived from statistical model. The details are provided in Lines 156-173.

The description of the four experiment groups are not clear in the method.

Response: We thank the reviewer for the comment. Based on the question raised by the reviewer, we believe that the confusion might be induced by the definitions of “forest management” and “improved management”. Therefore, we revised the text to make this clear to reader. Specifically, **forest management** (i.e. wood harvest, forest expansion) were represented for accurate projection of carbon dynamic trajectory. While in comparison, the **improved management practices** were further considered to quantify the additional carbon sink by adjusting management strategies, including the replacement of inappropriate tree species by indigenous species and reduction of the harvesting intensity by postponing the

harvesting time (see methods). We also revised the descriptions in the method section to clarify this information. Please check our revisions in Lines 78-92 and 441-458.

line 83, "without improved management"? in your simulations, harvest is included (Table S5). Is it not a type of management?

Response: We thank the reviewer for capturing this. Yes, harvest is a type of management, but in experiment groups 1, forest harvest was designed to follow the regular cycle (i.e. it is a management practice but not an improved management practice). The “improved management” of harvesting indicates that we extended the rotation length for additional carbon sink.

line 85, after "improved forest managements", add "i.e., expansion of new forest, wood harvest, age rotation, and tree replacement"

Response: We thank the reviewer for the suggestion. We added “i.e., wood harvest extension and tree replacement” in Line 88. This helps to clarify the meaning of “improved forest managements”. In this study, “**forest management**” includes the forest expansion and wood harvest, but “**improved forest managements**” specifically indicates the wood harvest extension and tree replacement practices which help further enhance carbon stock and sink (as listed in Table 1). In comparison, the optimal managements indicate the improved forest management practices portfolio achieving the largest carbon sink. We have revised the text in Lines 83-85 to make this clear to readers.

line 87, the 4th group, without considering any management?

Response: The 4th group experiments did consider forest management, such as wood harvesting. The fourth group was designed to keep forest area unchanged since 2020, i.e. no forest expansion. By comparing the 1st and the 4th experiment groups, we can quantify the contribution of existing and new forests to carbon sink. To ensure that the 1st and the 4th experiment groups comparable, forest management should be considered in both groups.

line 87-90, this definition should be ahead of division of the four groups.

Response: Done as suggested. Please see revisions in Lines 83-85.

line 94, what is LUH2?

Response: LUH2 is the Land-Use Harmonization (LUH2) dataset, and we have added this information in Line 98 for clarification.

line 117, why wood harvest is emphasized in the paper and other management measures (expansion of new forest age rotation, and tree replacement") are relatively less emphasized? Your simulations have included all of these measures, right?

Response: We thank the reviewer for pointing this out. As we clarified before, "forest management" includes the forest expansion and wood harvest, and "improved forest management" specifically indicates the wood harvest extension and tree replacement practices (as listed in Table 1). Therefore, wood harvest has been emphasized in experiment group 1 because it was the key process contributed to accurately capturing the carbon dynamic in forests. In comparison, the forest harvest age extension and tree replacement were "improved forest management" practices which was designed in experiment group 2 to quantify the additional carbon stock and sink from the improved management practices. Therefore, the forest harvest age extension and tree replacement were not mentioned until the last section of "Improved forest management strategies for forest carbon potential enhancement".

In summary, **the logic of the main text** was to first elaborate the carbon stock and sink estimation from group 1 experiments (by considering essential managements such as forest harvest), and then explain the results from group 2 experiments for exploring the additional sink from improved management strategies (by considering the improved practices such as tree replacement). During the elaboration of estimation from group 1 experiments, supports were needed to warrant and confirm their validity and accuracy (due to many improvements made in model simulations), therefore group 3 experiments were designed and used as a supplement for support.

line 138, "Specifically, the results of DLEM (i.e. process-based model) simulations were derived from group 1 experiments".

What does this sentence mean?

Response: We thank the reviewer for pointing this out. In Figure 1, the process-based simulation results were from group 1 experiments. As explained in the previous response, group 1 experiments were designed to give the results close to realistic under current management situation. Here we want to remind reader that the reported results were all derived from group 1 experiments, unless otherwise indicated.

The description of the results is not corresponding to the group settings. I am still confused when reading the results about which have and have not included (what type of) forest management.

Response: We thank the reviewer for providing this concern. We have made changes in the introduction section in response to previous concerns. Moreover, we further added the sentence "Note that the results reported were all derived from group 1 experiments, unless otherwise indicated." in the last paragraph of introduction to make this clear to reader. Please check our revision in lines 91-92. We hope these revisions in the introduction section (i.e. Lines 78-94) are competent.

line 138-142, is it to compare the results between with and without wood harvest? How did this result come from? "...primary of the sink (85%) is attributed to natural growth and regrowth of forests...".

Response: We found that this sentence can cause misunderstanding. Therefore, we deleted the sentence. As we explained before, the entire main text can be divided into two major sections, in which the first and second sections were derived from group 1 (i.e. under current forest management situation) and group 2 (i.e. under **improved** forest management scenarios) experiments, respectively. While in comparison, groups 3&4 experiments were only served as supplement to support the validity or factorial attributions of results in group 1. For example, the results in the sections before Line 299 were all derived from group 1 experiments, excepted for the results of group 3 (no harvest) in Lines 164-166 for supporting that DLEM simulations

were more accurate than statistical model (i.e. DLEM model yield almost the same estimation as statistical model if harvest was not considered).

line 145, "all scenarios with different tree survival rates and SSPs", from which scenarios?

Response: These include experiments S1-S8. We added this information in Line 152.

line 151, "Note that DLEM simulations were derived from experiments of simulation group 1 in Figure 1.

"What does this sentence mean? Other groups are also simulated by using DLEM.

Response: Here we want to remind that results were derived from group 1 experiments, because the results from group 3 were introduced later in this paragraph. To avoid confusion, we revised the text and please check our revisions in Lines 158-159.

The previous question is not answered properly. <> , your response is <<64 experiments were conducted>>. If so, this is not a "random" design. What I meant is, whether mortality (47% or 85%) is randomly designed for different cells across space.

Response: We thank the reviewer for the suggestion. Indeed, the experiments were not a "random" design. Instead, the simulations were designed to be constrained by the most possible ranges of assumptions, i.e. to capture the carbon dynamic in the forcing ranges close to realistic. Therefore, the survival rate (47% or 85%) was not randomly designed for cells, but it was set as a constrained assumption here. The other reason that the survival rate was not randomly designed for cells is because the computation load of the simulation is relatively high (e.g. 1-2 days per experiment), and randomization will greatly increase the experiment number (i.e. repeats) and is not currently affordable.

line 431, about validation, you stated in the response "the National Forest Inventory (NFI) survey data during 1999-2018 were used to calibrate and validate the model"? but in FigS14, validation was only using data from 2020. What about the other years?

Response: We thank the reviewer for the capturing this. This study focused on carbon dynamics in 2020-2100. Therefore, we purposely validated the carbon sink in the initial year of 2020. The

NFI data were available from 1999-2018, which were used to build the forest biomass growth equations for statistical model. The statistical model is advantageous in depicting the *status quo* of the carbon sink (but less capable in carbon dynamics projection as showed in this study). Therefore, in Figure S14, we used carbon sink derived from statistical model to validate process-based model at species level in 2020. We are sorry for the confusion. A more accurate statement is that "the National Forest Inventory (NFI) survey data during 1999-2018 were used to calibrate and validate the statistical model", and the estimation from the statistical model in 2020 was further used to validate the DLEM model. This is also showed in Line 520.

line 444, "wood harvest" is not included in "improved forest management"?

Response: We thank the reviewer for capturing this. As we clarified in the introduction section and also explained in this response letter, "wood harvest" is a "forest management" practice, but not included in "**improved** forest management" in this study.

line 450-451, group 1 and group 3 have different settings of CO₂. Therefore, the effect of CO₂ on carbon sequestration is not seperated from that of wood harvest. Please explain.

Response: We thank the reviewer for capturing this. Indeed, group 1 and group 3 have different settings of CO₂. Therefore, direct comparison between group 1 and group 3 is not appropriate (Lines 146-148 were deleted). Instead, results from group 3 were compared with statistical model (see description in Lines 164-166), because CO₂ effect was not included in statistical model. Moreover, as explained before, group 3 experiments were designed as a supplement to support that DLEM simulation was advantageous in capturing carbon dynamics over statistical model. In other words, our aim is to stress that, if the harvest is not considered, the process-based model (i.e. DLEM) could yield a carbon projection close to statistical model. Therefore, group 3 experiments were purposely designed to warrant the DLEM estimations. We further made this clear by revising the sentence in Lines 456-458. Note that results of group 3 experiments were only used in Lines 164-166 for comparison.

What is the idea of improve tree survival rate from 47% to 85%? How can it be achieved? How do you consider the relationship between tree survival rate and CO2 concentration?

Response: We thank the reviewer for pointing this out. We explained this in Lines 609-621. We made this assumption because planted forests might not always survive after planting, as forestation failure happens. We assumed that tree survival will be positioned between the harsh condition (e.g. arid and semi-arid regions) and the expected criteria designed by the State Forestry Administration (i.e. 85%). Therefore, for the period from 2021 onward, two scenarios were considered for forestation, in which the survival rates of the trees were 47% and 85%, respectively. The 47% tree survival rate was derived from the 9th NFI in the arid and semi-arid region of Northwest China, while 85% is the scenario assuming that the survival rate was improved by species-specific tending practices and land meets the criteria set by the State Forestry Administration. Thus, the survival rates were designed to force the model simulation close to realistic (i.e. to reflect the realistic conditions). We suspect that the higher survival rate can be achieved by species-specific tending practices (i.e. irrigation, fertilization, thinning). Unfortunately, these practices were site-dependent and are currently unable to be reflected in model simulations due to limited information.

The impacts of CO2 concentration on tree survival rate were not directly applied in model simulation due to lack of solid relationships available. Instead, as reported that tree mortality might increase due to higher climatic stresses and more frequent climatic extremes (e.g. pests, fires), we assumed that the tree mortality will increase from SSP126 to SSP585 in this study (see details in Lines 551-561).

FigS8, Is harvest included? In FigS10, the results represent scenarios of no harvest and age accrual. What about FigS8?

Response: Yes, harvest is a forest management practice and was considered in all experiments except group 3. Figure S8 was derived from group 1 experiments, while Figure S10 was derived from group 3 experiments. We have added this information in the figure titles for clarification. Please check our revisions in the titles of Figure S8&S10.

FigS11, what is the standard/principle for wood harvest setting between 2020 and 2100?

Response: The harvest setting was described in Lines 532-549. Concisely, we separated each tree species into timber forest and non-timber forest for both natural and planted forests. Harvesting was only performed on mature timber forest in simulations. The harvest age for each tree species was obtained from the State Forestry Administration of China (Table S6). The areas of timber forest for each species were obtained from the China Forest Resource Report 2014-2018. Besides, to ensure that forests were not overexploited, the total harvested forest was limited to the average annual allowable cut during a rotation period.

FigS12,13, not clear which scenario did the results represent?

Response: These figures were derived from group 1 experiments (i.e. averaged result of the S1-S8 experiments), and we have added this information in the figure title for clarification. Please check our revisions in the Figure S12&S13.

FigS15 How was the location of the new forest determined? what is the unit of the numbers in the figure?

Response: The new forest was determined by the forest expansion process described in Lines 609-630. In summary, the new forests were constructed following the National Forest Management Plan 2016-2050 and the “Mid- and long-term protection and recovery plan of natural forest in China (2022-2035)” released by the National Forestry and Grassland Administration of China (i.e. former National Forestry Administration). The officially released documents elaborated the areas of the new forest in each province. Besides, since China restricts further conversion of croplands and wetland to forestland for food security and biodiversity conservation, we assumed that the future plantations were only allowed to be converted from grassland and shrubland (maps obtained from <http://www/resdc.cn>). The distribution of the new forests was determined by the area, land availability map, and suitability index. Specifically, the grid-cell with higher suitability index for a specific species was given priority to be planted with the tree species (see details in Lines 632-654). The number in

Figure S15 indicates the coverage percentage of the forests. We have added this information in the Figure title. The source of the grassland and shrubland maps has been added in Lines 700-702.

FigS16 what is the unit of the numbers in the figure, possibility?

Response: The numbers indicate suitability index. We have added this information in the Figure title.

Table S5 56 experiments in group 2? What is the total number of all experiments?

Response: Yes, there are 56 experiments in group 2, because the N was set to 1, 2, or 3. Therefore, there were 56 experiments (i.e. 7 management scenarios \times 2 tree survival rate scenarios \times 4 SSPs). To quantify the contributions from existing and new forests, 8 experiments were added in group 4. The total number of all experiments is 74, including 8, 56, 2, and 8 experiments in group 1 to 4, respectively.

Reviewer #2 (Remarks to the Author):

Thank you for taking the time to address my concerns. This is a great study and I continue to be supportive of publication in Nature Communications.

Response: We thank the reviewer for valuing our work and the insightful suggestions that help improve the manuscript.

Reviewer #3 (Remarks to the Author):

I thanks the authors for their thorough revision of their manuscript, which, in my opinion, has improved a lot. The authors have answered to all my queries in a satisfactory way.

One thing though, regarding the "reduction of the range of uncertainties" (see my previous comment lines 104-106): I still think it would be more stringent to rather speak of this as an "exploration of realistic sensitivity analysis based on Chinese forestry plan with data-constrained by the 9th NFI

integrated into validated models".

But despite this last comment, I can only congratulate the authors for their work on revising the manuscript.

Response: We thank the reviewer for the suggestion. We agree and have revised the descriptions of "reducing uncertainty" by replacing the term with "enhance the accuracy of estimates" as suggested in the text. Please check our revisions in Lines 105-106, 141-143.